# Intrinsic Sliced Wasserstein Distances for Comparing Collections of Probability Distributions on Manifolds and Graphs

## Abstract

Collections of probability distributions arise in a variety of statistical applications ranging from user activity pattern analysis to brain connectomics. In practice these distributions are represented by histograms over diverse domain types including finite intervals, circles, cylinders, spheres, other manifolds, and graphs. This paper introduces an approach for detecting differences between two collections of histograms over such general domains. We propose the intrinsic slicing construction that yields a novel class of Wasserstein distances on manifolds and graphs. These distances are Hilbert embeddable, allowing us to reduce the histogram collection comparison problem to a more familiar mean testing problem in a Hilbert space. We provide two testing procedures, one based on resampling and another on combining $p$-values from coordinate-wise tests. Our experiments in a variety of data settings show that the resulting tests are powerful and the $p$-values are well-calibrated. Example applications to user activity patterns and spatial data are provided.

## 1 Introduction

Distributional data arise in a variety of statistical applications. In practice these are not limited to distributions over real intervals, but are often defined over manifolds and graphs. For instance, even in the simplest case of analyzing 24-hour activity patterns by constructing histograms of activity counts by time, the resulting histograms are really supported on a circle rather than an interval on the real line. If in addition to the time of activity, the observations come with a real number such as the intensity of the activity, then we end up with a histogram over a cylindrical domain. Spatial datasets recorded at some geographic region level are another example: one can build a distribution over the region adjacency graph by capturing the normalized counts of events in each region. When analyzing distributions over such general domains it is desirable to rely on methods that take into account the connectivity and geometry of the underlying domain, respect the distributional nature of the data, and lead to efficient practical algorithms.

In this paper we consider the problem of comparing two collections of distributions, namely testing for homogeneity—whether all of the distributions come from the same *meta-distribution*. While conceptually similar to two-sample testing, this is a higher order notion in the sense that our units of analysis are distributions/histograms. Letting $\mathcal{P}(\mathcal{X})$ denote the set of Borel probability measures on a metric space $\mathcal{X}$, consider the space $\mathcal{P}(\mathcal{P}(\mathcal{X}))$. Let $P, Q \in \mathcal{P}(\mathcal{P}(\mathcal{X}))$, and assume that we are given two collections of probability measures $\{\mu_i\}_{i=1}^{N_1}$ and $\{\nu_i\}_{i=1}^{N_2}$ that are drawn from $P$ and $Q$, $\mu_i \sim P$ and $\nu_i \sim Q$ in an independent and identical manner. Our goal is to test whether $P = Q$ and, moreover, to be able to conduct such tests for general $\mathcal{X}$. Such a test would be useful in numerous practical situations. For instance, an online retailer may aggregate a customer's monthly activity

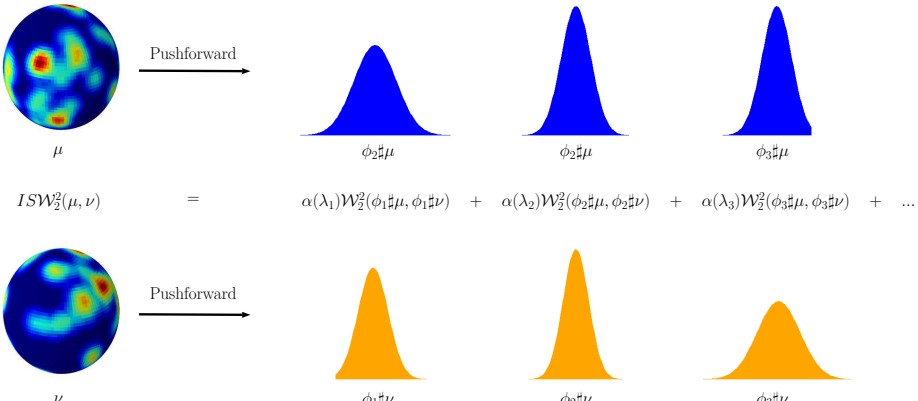

Figure 1: Schematic of the proposed intrinsic slicing construction. Given two probability measures on the sphere (here the darkest blue corresponds to zero mass), different aspects of their dissimilarities become apparent after pushforward to the real line using the eigenfunctions of the Laplace-Beltrami operator, $\{\phi_i\}$, in this case spherical harmonics. As a particular example of our general construction, the (squared) intrinsic sliced 2-Wasserstein distance $ISW_2^2(\cdot, \cdot)$ is the weighted sum of the dissimilarities of the corresponding pushforwards of $\mu$ and $\nu$ as measured by squared 2-Wasserstein distance $\mathcal{W}_2^2(\cdot, \cdot)$ *on the real line.*

into a histogram over a cylinder capturing the time of the day and amount of purchase for each transaction. By considering collections of histograms for various customer segments, one can conduct tests to determine if there are statistically significant differences between behavioral patterns of these segments.

We attack this problem using insights from recent developments that utilize *Hilbert embeddings* for simplifying distributional data problems (see e.g. [18, 24] for particular examples). The simplification comes as a result of linearity of Hilbert spaces, which allows adapting existing statistical approaches such as functional data methodology to distributional data. A crucial requirement on the embedding is that the distance in the embedding space should give a meaningful distance between measures; it is this property that renders quantities computed in the embedding space such as means and variances meaningful. Thus, embedding constructions should be driven by specifying appropriate distances on the space of measures. Of course, not every distance can be embedded and Hilbert embeddable distances are called Hilbertian; see [19] for an overview of this notion.

The focus in this paper will be on transportation based distances between distributions/histograms [25]. Other approaches such as bin-wise treatment of histograms may result in increased variability when horizontal variation is present, leading eventually to less powerful methods. Transportation based distances are more efficient at capturing this and other aspects of distributional data [3, 17, 19]. However, adopting the transportation theoretic approaches to our problem immediately hits a roadblock beyond the real line case: while 2-Wasserstein distance on the real line is Hilbert embeddable, it fails to be so on general domains [19]. The general Hilbert embedding framework of [18] is not tied to a distance between probability distributions and so can be problematic for capturing the location and variability aspects of distribution collections. In addition, [18] has difficulties in higher dimensions and does not provide constructions suitable for manifolds or graphs.

Inspired by the sliced 2-Wasserstein distances in high dimensional spaces [11, 12], we introduce a new slicing construction (Figure 1) that leverages the eigenvalues and eigenfunctions of the Laplace-Beltrami operator on manifolds and graph Laplacians to capture the intrinsic geometry and connectivity of the domain. We apply this slicing construction to obtain a novel class of intrinsic sliced 2-Wasserstein distances on manifolds and graphs. The resulting distances are Hilbert embeddable, have a number of desirable properties, and can be truncated to obtain finite-dimensional embeddings. Using the corresponding embedding allows us to reduce the histogram collection comparison problem to the comparison of means in a high-dimensional Euclidean space. We provide two approaches for hypothesis testing and verify via extensive experiments on synthetic and real data examples in a variety of data settings that these tests are powerful, and the $p$-values are well-calibrated.

Comparing with closely related work, while the Generalized Sliced Wasserstein (GSW) distance [11] sets up the idea of approximating Wasserstein distances using multiple nonlinear projections, it is presented in extrinsic terms (i.e. Euclidean space) and can suffer from the curse of dimensionality when a low dimensional data manifold lives in a high-dimensional space. Our choice of eigenfunctions for projection is very different from the one-parameter function families in GSW. Moreover, the GSW construction does not directly apply to graphs. While the tree-sliced variant of GSW [14] can be applied in an intrinsic manner (the clustering variant), it relies on a different type of distance, in the limit related to the euclidean/geodesic distance. This can be seen by comparing our lower bound to theirs: our lower bound for ISW is in terms of the MMD using the spectral distance (Proposition 5). Finally, the robust sliced Wasserstein distance of [13] does make use of the geometric properties of the underlying manifold. However, their goal is to compute a correspondence between two manifolds by mapping them into $\mathbb{R}^d$ using eigenmaps and treating the mapped manifolds as measures in $\mathbb{R}^d$ and minimizing some version of Euclidean slicing.

## 2 Preliminaries

Given a compact metric space $\mathcal{X}$, let $\mathcal{P}(\mathcal{X})$ denote the set of Borel probability measures on $\mathcal{X}$. Our main interest is in the case where $\mathcal{X}$ is a graph or a manifold with the shortest/geodesic distance as the metric, and thus the compactness restriction. The 2-Wasserstein distance can be defined on $\mathcal{P}(\mathcal{X})$ using the metric of $\mathcal{X}$ as the ground distance [17, 19], giving $\mathcal{W}_2^{\mathcal{X}} : \mathcal{P}(\mathcal{X}) \times \mathcal{P}(\mathcal{X}) \to \mathbb{R}_{\geq 0}$; due to the repeated use of the real line case we use the shorthand $\mathcal{W}_2 = \mathcal{W}_2^{\mathbb{R}}$. Central to our study are distributions on the space of probability measures $\mathcal{P}(\mathcal{P}(\mathcal{X})) = (\mathcal{P}(\mathcal{X}), \mathcal{B}(\mathcal{P}(\mathcal{X})))$, where $\mathcal{B}(\mathcal{P}(X))$ is the Borel $\sigma$-algebra generated by the topology induced by $\mathcal{W}_2^{\mathcal{X}}$ [3]. To avoid confusion, we will refer to the elements of $\mathcal{P}(\mathcal{P}(\mathcal{X}))$ as *meta-distributions*.

Let $P, Q \in \mathcal{P}(\mathcal{P}(\mathcal{X}))$, and assume that we are given two collections of probability measures $\{\mu_i\}_{i=1}^{N_1}$ and $\{\nu_i\}_{i=1}^{N_2}$ that are drawn from $P$ and $Q$: $\mu_i \sim P$ and $\nu_i \sim Q$ in an independent-and-identically-distributed (hereafter i.i.d.) manner. Our goal is to use this sample to test the null hypothesis of whether $P = Q$. While this is conceptually a two-sample test, note that our data points are distributions; in practice, the distributions $\mu_i$ or $\nu_i$ are given by histograms.

*Remark* 1. Let us compare this with the usual two-sample testing. Consider $P \in \mathcal{P}(\mathcal{P}(\mathcal{X}))$ constructed as follows. Let $\mu^* \in \mathcal{P}(\mathcal{X})$ be a fixed probability measure. Let $x_1, x_2, ... x_A \sim \mu^*$ and construct the histogram summarizing this sample: $\frac{1}{A} \sum_{a=1}^{A} \delta_{x_a}$. Now, $\frac{1}{A} \sum_{a=1}^{A} \delta_{x_a} \in \mathcal{P}(\mathcal{X})$ is one sample drawn from $P$. In our testing scenario one gets the collection $\{\mu_i\}_{i=1}^{N_1}$, where each histogram is obtained as above: $\mu_i \sim P$. Similarly, consider $Q \in \mathcal{P}(\mathcal{P}(\mathcal{X}))$ of the same type based on some other fixed $\nu^* \in \mathcal{P}(\mathcal{X})$, and let $\{\nu_i\}_{i=1}^{N_2}$ the corresponding collection of histograms. Testing whether $P = Q$ in the limit boils down to $\mu^* = \nu^*$. When compared to the usual two-sample testing this may seem rather inefficient, requiring $A$ times more samples (resp. $N_1 A$ and $N_2 A$ samples from $\mu^*$ and $\nu^*$). However, in our setup it is *not assumed* that the histograms in the collections come from meta-distributions of the above simple type (i.e. all $\mu_i$ are generated by drawing from the same underlying distribution $\mu^*$). In fact, the target use-case for our approach is when these histograms are collected by observing different individuals who have their *person-specific* behaviors/distributions. □

Let $\mathcal{D}(\cdot, \cdot) : \mathcal{P}(\mathcal{X}) \times \mathcal{P}(\mathcal{X}) \to \mathbb{R}_{\geq 0}$ be a distance between probability distributions. $\mathcal{D}(\cdot, \cdot)$ is called *Hilbertian* (this is just a naming convention; no implication that the map is a Hilbert map) if there exist a Hilbert space $\mathcal{H}$ and a map $\eta : \mathcal{P}(\mathcal{X}) \to \mathcal{H}$ such that $\mathcal{D}(\mu, \nu) = \|\eta(\mu) - \eta(\nu)\|_{\mathcal{H}}$. For example, it is well-known that 2-Wasserstein distance on $\mathcal{X} = \mathbb{R}$ is Hilbertian [19] (also see Section 3.2) and Maximum Mean Discrepancy (MMD) on any $\mathcal{X}$ is Hilbertian [8]; however, the 2-Wasserstein distance $\mathcal{W}_2^{\mathcal{X}}$ on general $\mathcal{X}$ is not Hilbertian [19].

Since the map $\eta$ takes every measure on $\mathcal{X}$ to a point in $\mathcal{H}$, we see that a process $P \in \mathcal{P}(\mathcal{P}(\mathcal{X}))$ gives a rise to a measure on $\mathcal{H}$ given by pushforward operation, $\eta \# P = P \circ \eta^{-1} \in \mathcal{P}(\mathcal{H})$. In addition, if a finite dimensional approximation $\eta_D : \mathcal{P}(\mathcal{X}) \to \mathbb{R}^D$ of $\eta$ is available, then $\eta_D \# P$ is a measure on $\mathbb{R}^D$. This observation is enormously useful: problems about the elements of the rather abstract space $\mathcal{P}(\mathcal{P}(\mathcal{X}))$ are reduced to problems about familiar measures on $\mathcal{H}$ or even $\mathbb{R}^D$. For example, the usual notions of mean and variance can be applied to the measure $\eta \# P$ to gain insights about the meta-distribution $P$. The validity of these insights hinges on the $\eta$-map coming from a Hilbertian distance, as distances are central to the statistical quantities of interest.

Testing for $\eta\#P = \eta\#Q$ can serve as a proxy for our original testing problem of $P = Q$. As typical with two-sample tests, various aspects of the equality $\eta\#P = \eta\#Q$ can be tested, such as the mean or variance equality; unspecific tests of equality can be applied as well. We will concentrate on testing certain aspects of the equality so that one can easily drill down on the results. This is similar to the regular two-sample testing where checking for equality of, say, means is often preferrable as it gives immediately interpretable insights, whereas a general test that only says there are unspecified differences between the distributions is less useful for interpretation. To obtain succint and interpretable tests we concentrate on the mean of the resulting pushforward measure in $\mathcal{H}$.

**Definition 1.** For a meta-distribution $P \in \mathcal{P}(\mathcal{P}(\mathcal{X}))$, we define its *Hilbert centroid* with respect to the Hilbertian distance $\mathcal{D}$ as $C_{\eta\#P} = \mathbb{E}_{\mu\sim P}[\eta(\mu)] \in \mathcal{H}$, assuming it exists.

Our testing procedure is based on checking the equality $C_{\eta\#P} = C_{\eta\#Q}$, or more explicitly: $\mathbb{E}_{\mu\sim P}[\eta(\mu)] = \mathbb{E}_{\nu\sim Q}[\eta(\nu)]$. Intuitively, each "dimension" of the map $\eta$ probes some aspect of the two involved meta-distributions and makes sure that they are in agreement in expectation. One of our testing approaches will use the statistic

$$\mathbb{T}(P,Q) = \|C_{\eta\#P} - C_{\eta\#Q}\|_{\mathcal{H}}^2. \tag{2.1}$$

to capture the deviations from equality; this quantity can be written directly in terms of pairwise distances.

**Proposition 1.** *For $P, Q \in \mathcal{P}(\mathcal{P}(\mathcal{X}))$, the following holds:*

$$\mathbb{T}(P,Q) = \mathbb{E}_{\mu\sim P, \nu\sim Q}[\mathcal{D}^2(\mu,\nu)] - \frac{1}{2}\mathbb{E}_{\mu,\mu'\sim P}[\mathcal{D}^2(\mu,\mu')] - \frac{1}{2}\mathbb{E}_{\nu,\nu'\sim Q}[\mathcal{D}^2(\nu,\nu')].$$

Next we give an example of what Hilbert centroid equality means in an important special case.

**Example 1.** Let $\mathcal{X} = [0, T] \subset \mathbb{R}$ with $\mathcal{D}$ being the 2-Wasserstein distance $\mathcal{W}_2$. Given a probability measure $\mu \in \mathcal{P}([0,T])$, let $F_\mu$ be its cumulative distribution function: $F_\mu(x) = \mu([0,x]) = \int_0^x d\mu$. The generalized inverse of cumulative distribution function (CDF) is defined by $F_\mu^{-1}(s) := \inf\{x \in [0,T] : F_\mu(x) > s\}$. The squared 2-Wasserstein distance has a rather simple expression in terms of the inverse CDF [19]:

$$\mathcal{W}_2^2(\mu,\nu) = \int_0^1 (F_\mu^{-1}(s) - F_\nu^{-1}(s))^2 ds. \tag{2.2}$$

This formula immediately establishes the Hilbertianity of $\mathcal{W}_2$ through the map $\eta : \mathcal{P}([0,T]) \to L_2([0,T])$ defined by $\eta(\mu) = F_\mu^{-1}$. Note that $\eta$ is invertible for increasing normalized functions in the embedding space. Using this insight, we see that the corresponding "average measure" of $P \in \mathcal{P}(\mathcal{P}(\mathcal{X}))$ can be introduced via $P_{\mathrm{av}} = \eta^{-1}(\mathbb{E}_{\mu\sim P}[\eta(\mu)])$. It is easy to prove that $P_{\mathrm{av}}$ satisfies the following: $P_{\mathrm{av}} = \arg\min_{\rho\in\mathcal{P}(\mathcal{X})} \mathbb{E}_{\mu\sim P}[\mathcal{W}_2(\mu,\rho)^2]$, which is the definition of the Fréchet mean, see for example [19]. In this setting, $C_{\eta\#P} = C_{\eta\#Q}$ boils down to having the same Fréchet means, $P_{\mathrm{av}} = Q_{\mathrm{av}}$. $\qquad\square$

We will later see that the Hilbert embedding corresponding to the intrinsic sliced 2-Wasserstein distance is assembled of embeddings like in Example 1 applied after pushforwards (see Figure 1 for an intuition). This means that the resulting equality $C_{\eta\#P} = C_{\eta\#Q}$ becomes more stringent, making it a better proxy for detecting the deviations from $P = Q$ without losing the interpretability aspect.

## 3 Intrinsic Sliced 2-Wasserstein Distance

We introduce a Hilbertian version of $\mathcal{W}_2$ on manifolds and graphs via a construction we call *intrinsic slicing* due to its use of the domain's intrinsic geometric properties. To focus our discussion we concentrate on the manifold case, as the graph case is simpler and is obtained by replacing the Laplace-Beltrami operator by the graph Laplacian.

Let $\lambda_\ell, \phi_\ell; \ell = 0, 1, \dots$ be the eigenvalues and eigenfunctions of the Laplace-Beltrami operator on $\mathcal{X}$ with Neumann boundary conditions. The eigenfunctions are sorted by increasing eignevalue and assumed to be orthonormal with respect to some fixed (e.g. uniform) measure on $\mathcal{X}$; also $\phi_0 = const$ and $\lambda_0 = 0$. One can define the spectral kernel $k(x,y) = \sum_\ell \alpha(\lambda_\ell)\phi_\ell(x)\phi_\ell(y)$ and the corresponding spectral distance on the manifold $d(x,y) = k(x,x) + k(y,y) - 2k(x,y) = \sum \alpha(\lambda_\ell)(\phi_\ell(x) - \phi_\ell(y))^2$, where $\alpha : \mathbb{R}_{\geq 0} \to \mathbb{R}_{\geq 0}$ is a function that controls contribution from

each spectral band. By setting $\alpha(\lambda) = e^{-t\lambda}$ for some $t > 0$, we get the heat/diffusion kernel and the corresponding diffusion distance [4]. Another important case is $\alpha(\lambda) = 1/\lambda^2$ if $\lambda > 0$ and $\alpha(0) = 0$, which gives the biharmonic kernel and distance [15]. In both of these constructions $\alpha(\cdot)$ is a decreasing function, allowing the smoother low-frequency (i.e. smaller $\lambda_\ell$) eigenfunctions to contribute more.

## 3.1 Definition and properties

A real-valued function $\phi : \mathcal{X} \to \mathbb{R}$ can be used to map the manifold $\mathcal{X}$ onto the real line. Any probability measure $\mu \in \mathcal{P}(\mathcal{X})$ can likewise be projected onto the real line using the pushforward of $\phi$, which we denote by $\phi\sharp\mu = \mu \circ \phi^{-1} \in \mathcal{P}(\mathbb{R})$. While the pushforward notions used here and in previous sections are conceptually the same, for clarity we use $\sharp$ for measures and $\#$ for meta-distributions. We define intrinsic slicing as follows.

**Definition 2.** Given a function $\alpha : \mathbb{R}_{\geq 0} \to \mathbb{R}_{\geq 0}$ and a probability distance $\mathcal{D}(\cdot, \cdot)$ on $\mathcal{P}(\mathbb{R})$, we define the intrinsic sliced distance $IS\mathcal{D}(\cdot, \cdot)$ on $\overline{\mathcal{P}}(\mathcal{X})$ by

$$IS\mathcal{D}^2(\mu, \nu) = \sum_\ell \alpha(\lambda_\ell)\mathcal{D}^2(\phi_\ell\sharp\mu, \phi_\ell\sharp\nu).$$

The choice of the Laplacian eigenfunctions in the definition can be justified by a number of their properties. Eigenfunctions are intrinsic quantities of a manifold and are ordered by smoothness. Thus, they allow capturing the intrinsic connectivity of the underlying domain. Furthermore, due to the orthogonality of eigenfunctions, their pushforwards can capture complementary aspects.

While the definition is general, our focus in this paper is on the case when $\mathcal{D} = \mathcal{W}_2$; we remind that we always use $\mathcal{W}_2$ to denote the 2-Wasserstein distance on $\mathcal{P}(\mathbb{R})$. We call the resulting distance *Intrinsic Sliced 2-Wasserstein Distance*, and denote it by $IS\mathcal{W}_2$. First, we discuss the convergence of the infinite sum in Definition 2.

**Proposition 2.** *If $\mathcal{X}$ is a smooth compact $n$-dimensional manifold and $\sum_\ell \lambda_\ell^{(n-1)/2}\alpha(\lambda_\ell) < \infty$, then $IS\mathcal{W}_2$ is well-defined.*

Next, we prove a number of properties of $IS\mathcal{D}$.

**Proposition 3.** *If $\mathcal{D}$ is a Hilbertian probability distance such that $IS\mathcal{D}$ is well-defined, then (i) $IS\mathcal{D}$ is Hilbertian, and (ii) $IS\mathcal{D}$ satisfies the following metric properties: non-negativity, symmetry, the triangle inequality, and $IS\mathcal{D}(\mu, \mu) = 0$.*

*Proof.* By Hilbertian property of $\mathcal{D}$, there exists a Hilbert space $\mathcal{H}^0$ and a map $\eta^0 : \mathcal{P}(\mathbb{R}) \to \mathcal{H}^0$ such that $\mathcal{D}(\rho_1, \rho_2) = \|\eta^0(\rho_1) - \eta^0(\rho_2)\|_{\mathcal{H}'}$ for all $\rho_1, \rho_2 \in \mathcal{P}(\mathbb{R})$. Plugging this into Definition 2 we have $IS\mathcal{D}(\mu, \nu) = \|\eta(\mu) - \eta(\nu)\|_{\mathcal{H}}$, where $\mathcal{H} = \oplus_\ell \mathcal{H}^0$ and the $\ell$-th component of $\eta(\mu)$ is $\sqrt{\alpha(\lambda_\ell)}\eta_0(\phi_\ell\sharp\mu) \in \mathcal{H}$. The second part of Proposition 3 directly follows from the Hilbert property. $\square$

Since $\mathcal{W}_2$ is Hilbertian on $\mathcal{P}(\mathbb{R})$, the application of Proposition 3 yields that $IS\mathcal{W}_2$ is also Hilberitan, making it possible to use $IS\mathcal{W}_2$ for our hypothesis tests in Section 4.

The following result shows that $IS\mathcal{W}_2$ inherits an important property of the Wasserstein distances, namely that the distance between two Dirac delta measures equals to a specific ground distance between their locations.

**Proposition 4.** *When $\mu = \delta_x(\cdot), \nu = \delta_y(\cdot)$ for two points $x, y \in \mathcal{X}$, we have $IS\mathcal{W}_2(\mu, \nu) = d(x, y)$, where $d(\cdot, \cdot)$ is the spectral distance corresponding to the choice of $\alpha(\cdot)$.*

For a simple choice of distance $\mathcal{D}$ on $\mathcal{P}(\mathbb{R})$, namely the absolute mean difference, the corresponding intrinsic sliced distance is the well-known MMD [8].

**Proposition 5.** *Let $\mathcal{D}(\rho_1, \rho_2) = |\mathbb{E}_{x\sim\rho_1}[x] - \mathbb{E}_{y\sim\rho_2}[y]|$ for $\rho_1, \rho_2 \in \mathcal{P}(\mathbb{R})$, then the corresponding $IS\mathcal{D}$ is equivalent to the MMD with the spectral kernel $k(\cdot, \cdot)$.*

When $k(x, y)$ is the heat kernel, the sliced distance in Proposition 5 is very much like the MMD with the Gaussian kernel, with the parameter $t$ in $\alpha(\lambda) = e^{-t\lambda}$ controlling the kernel width. Indeed,

the two kernels coincide on $\mathbb{R}^d$, and on general manifolds Varadhan's formula gives asymptotic equivalence for small $t$ [2].

An interesting insight derived from the above result is that $ISW_2$ is in a sense a "stronger" distance than MMD that uses the corresponding spectral kernel. The $ISW_2$ compares the quantiles of pushforward distributions (Eq. (2.2)), whereas MMD compares their expectations only. We formalize this notion next, also providing a theoretical reason for preferring $ISW_2$ for hypothesis testing.

**Proposition 6.** $MMD(\mu, \nu) \leq ISW_2(\mu, \nu)$ *when the same* $\alpha(\cdot)$ *is used in both constructions.*

*Proof.* This follows directly from the fact that for $\rho_1, \rho_2 \in \mathcal{P}(\mathbb{R})$ the inequality $|\mathbb{E}_{x \sim \rho_1}[x] - \mathbb{E}_{y \sim \rho_2}[y]| \leq \mathcal{W}_2(\rho_1, \rho_2)$ holds. $\qquad\square$

We are now in a position to prove that $ISW_2$ is a true metric.

**Theorem 1.** *If* $\alpha(\lambda) > 0$ *for all* $\lambda > 0$ *, then* $ISW_2$ *is a metric on* $\mathcal{P}(\mathcal{X})$.

We remind that 2-Wasserstein distance can be defined directly on $\mathcal{P}(\mathcal{X})$ using the geodesic distance as the ground metric; we denote this distance as $\mathcal{W}_2^{\mathcal{X}}$. Lipschitz properties of the eigenfunctions imply the following:

**Proposition 7.** *There exists a constant* $c$ *depending only on* $\mathcal{X} \subseteq \mathbb{R}^n$ *such that for all* $\mu, \nu \in \mathcal{P}(\mathcal{X})$ *the inequality* $ISW_2(\mu, \nu) \leq c\mathcal{W}_2^{\mathcal{X}}(\mu, \nu)\sqrt{\sum_\ell \lambda_\ell^{(n+3)/2}\alpha(\lambda_\ell)}$ *holds.*

Our final result looks at the quantity $\mathbb{T}$ defined using $ISW_2$ by Eq. (2.1). We will be using $\mathbb{T}$ computed on finite collections of measures as a test statistic in the next section. We show that it enjoys robustness with respect to small perturbations of the measures in the collection.

**Proposition 8.** *Let* $\{\mu_i\}_{i=1}^N$ *and* $\{\nu_i\}_{i=1}^N$ *be two collections of probability measures on* $\mathcal{P}(\mathcal{X})$, *such that* $\forall i, \mathcal{W}_2^{\mathcal{X}}(\mu_i, \nu_i) \leq \epsilon$, *then* $\mathbb{T}(\{\mu_i\}_{i=1}^N, \{\nu_i\}_{i=1}^N) \leq C^2\epsilon^2$. *Here* $C = c\sqrt{\sum_\ell \lambda_\ell^{(n+3)/2}\alpha(\lambda_\ell)}$ *from previous proposition and is assumed to be finite.*

This bound implies that if the distributions in a collection undergo horizontal shifts that are small as measured by the geodesic Wasserstein distance $\mathcal{W}_2^{\mathcal{X}}$, then $\mathbb{T}$ is small as well.

## 3.2 Approximate Hilbert Embedding

An important aspect of $ISW_2$ is that its Hilbert map $\eta : \mathcal{P}(\mathcal{X}) \to \mathcal{H}$ can be approximated by a finite-dimensional embedding $\eta_D : \mathcal{P}(\mathcal{X}) \to \mathbb{R}^D$ such that $ISW_2(\mu, \nu) \approx \|\eta_D(\mu) - \eta_D(\nu)\|_{\mathbb{R}^D}$. This is useful for practical computation and for one of our hypothesis testing approaches.

Using the formula for $ISW_2$ on $\mathcal{P}(\mathbb{R})$ in terms of the quantile function, Eq. (2.2), the Hilbert map is defined by $\eta^0(\mu) = F_\mu^{-1}$. We have $\mathcal{W}_2(\mu, \nu) = \|\eta^0(\mu) - \eta^0(\nu)\|_{L_2(\mathbb{R})}$, where the norm involves integration. We can discretize the integral using the Riemann sum for equidistant knots $s_k = \frac{k-1}{D'}, k = 1, ..., D'$, define the approximate embedding $\eta_{D'}^0 : \mathcal{P}(\mathbb{R}) \to \mathbb{R}^{D'}$ as:

$$\eta_{D'}^0 : \mu \to \frac{1}{\sqrt{D'}}[F_\mu^{-1}(s_1), ..., F_\mu^{-1}(s_{D'})]. \tag{3.1}$$

Now, $\mathcal{W}_2(\mu, \nu) \approx \|\eta_{D'}^0(\mu) - \eta_{D'}^0(\nu)\|_{\mathbb{R}^{D'}}$ with approximation quality depending on the embedding dimension $D'$.

To approximate the Hilbert map for $ISW_2$ we truncate the series defining $ISW_2$ and use a finite number of eigenfunctions for pushforward: $\phi_\ell, \ell = 1, ..., L$, where $\phi_0$ is dropped since it is a constant. By inspecting the proof of Proposition 3 and using Eq. (3.1), we can define $\eta_D : \mathcal{P}(\mathcal{X}) \to \mathbb{R}^D$ with $D = LD'$ as the concatenation of $L$ maps:

$$(\eta_D)_\ell : \mu \to \sqrt{\frac{\alpha(\lambda_\ell)}{D'}}[F_{\phi_\ell \sharp \mu}^{-1}(s_1), ..., F_{\phi_\ell \sharp \mu}^{-1}(s_{D'})].$$

Spectral decompositions of the Laplace-Beltrami operators for general manifolds [4, 20] or graph Laplacians can be computed numerically. For applications that involve simple manifolds, the eigenvalues and eigenfunctions can be computed analytically (see Appendix).

## 4 Hypothesis Testing

Let $\{\mu_i\}_{i=1}^{N_1}$ and $\{\nu_i\}_{i=1}^{N_2}$ be two i.i.d. collections of measures drawn from $P, Q \in \mathcal{P}(\mathcal{P}(\mathcal{X}))$ respectively. Our goal is to use these samples to test the null hypothesis $H_0 : C_{\eta\#P} = C_{\eta\#Q}$, where $\eta$ is the Hilbert embedding of the sliced distance $ISW_2$ on $\mathcal{P}(\mathcal{X})$.

### 4.1 Resampling Based Test

We use the quantity $\mathbb{T}(\cdot, \cdot)$ from Eq. (2.1) as the test statistic. Its sample version is computed by replacing the expectations by the empirical means, and excluding the diagonal terms to achieve unbiasedness

$$\hat{\mathbb{T}} \equiv \sum_{i,j:i\neq j} \frac{ISW_2^2(\mu_i, \mu_j)}{2N_1(N_1 - 1)} + \sum_{i,j:i\neq j} \frac{ISW_2^2(\nu_i, \nu_j)}{2N_2(N_2 - 1)} - \sum_{i,j} \frac{ISW_2^2(\mu_i, \nu_j)}{N_1 N_2}.$$

Note that $\mathbb{E}\hat{\mathbb{T}} = \mathbb{T}(P, Q)$. In practice, the $ISW_2$ values are computed from the approximate embedding: $ISW_2(\rho_1, \rho_2) \approx \|\eta_D(\rho_1) - \eta_D(\rho_2)\|_{\mathbb{R}^D}$. We denote the resulting statistic by $\tilde{\mathbb{T}}_{L,D'}$.

The difference between $\tilde{\mathbb{T}}_{L,D'}$ and the population version (i.e. $\mathbb{T} - \tilde{\mathbb{T}}_{L,D'}$) can be decomposed as $(\mathbb{T} - \hat{\mathbb{T}}) + (\hat{\mathbb{T}} - \hat{\mathbb{T}}_L) + (\hat{\mathbb{T}}_L - \tilde{\mathbb{T}}_{L,D'})$, where the summands inside the terms $\hat{\mathbb{T}}_L$ and $\tilde{\mathbb{T}}_{L,D'}$ correspond to partial sums that approximate $ISW_2^2(\cdot, \cdot)$ by $\sum_{l=1}^{L} \alpha(\lambda_l)\mathcal{W}_2^2(\phi_l\sharp\cdot, \phi_l\sharp\cdot)$, and $\mathcal{W}_2^2(\phi_l\sharp\cdot, \phi_l\sharp\cdot)$ by $\|\eta_{D'}(\phi_l\sharp\cdot) - \eta_{D'}(\phi_l\sharp\cdot)\|^2$, respectively. We show in Appendix that a) summands in the second and third terms in the sum can be made infinitesmally small by choosing large enough $L$ and $D'$, respectively; b) an asymptotic result for the first difference can be obtained by extending the tools from [8, 23]. These results are based on several assumptions detailed in the Appendix. Combining the two results, we establish asymptotic distributions of $\tilde{\mathbb{T}}_{L,D'}$:

**Theorem 2.** *Assume relevant conditions (see Appendix) hold. Define $N = N_1 + N_2$, and suppose that as $N_1, N_2 \to \infty$, we have $N_1/N \to \rho_1, N_2/N \to \rho_2 = 1 - \rho_1$, for some fixed $0 < \rho_1 < 1$. With $L \geq L_N, D' \geq D_N$ chosen in an appropriate way (see Appendix), under $H_0 : C_{\eta\#P} = C_{\eta\#Q}$ we have*

$$N\tilde{\mathbb{T}}_{L,D'} \rightsquigarrow \sum_{m=1}^{\infty} \gamma_m(A_m^2 - 1),$$

*where $A_m \sim N(0,1)$ for $m = 1, 2, \ldots$, and $\gamma_m$ are the eigenvalues of a certain operator that depends on $P$ and $Q$. Further, under $H_1 : C_{\eta\#P} \neq C_{\eta\#Q}$, $\sqrt{N}\left(\tilde{\mathbb{T}}_{L,D'} - \mathbb{T}\right)$ is asymptotically Gaussian with mean 0 and finite variance.*

We evaluate the power performance of the testing procedure based on $\tilde{\mathbb{T}}_{L,D'}$ for the sequence of contiguous alternatives $H_{1N} = \{(P, Q) : C_{\mu\#P} = C_{\mu\#Q} + \delta_N, l = 1, 2, \ldots\}$, where the deviation from null is quantified collectively by pushforward differences $\delta_{\ell N} \in \mathcal{H}, \delta_N = \oplus_\ell(\sqrt{\alpha_\ell}\delta_{\ell N})$ that are made to approach 0 as $N \to \infty$. The following theorem establishes consistency of our testing procedure against a family of such local alternatives.

**Theorem 3.** *Assume conditions (i)-(iii) hold, and let $L, D'$ be chosen as in Theorem 2. Then for the sequence of contiguous alternatives $H_{1N}$ such that $N\|\delta_N\|_{\mathcal{H}^*}^2 \to \infty$, the test based on $\tilde{\mathbb{T}}_{L,D'}$ is consistent for any $\alpha \in (0,1)$, that is as $N \to \infty$ the asymptotic power approaches 1.*

**Testing Procedure** In practice, to obtain the $p$-value for the $\tilde{\mathbb{T}}_{L,D'}$-statistic we use a bootstrap procedure. Remember that $\tilde{\mathbb{T}}_{L,D'}$ is computed via the approximate embedding $\eta_D$ with $D = LD'$. The collection $\{\mu_i\}_{i=1}^{N_1}$ is mapped to the collection $\{X_i = \eta_D(\mu_i)\}_{i=1}^{N_1}$ of vectors in $\mathbb{R}^D$ drawn in an i.i.d. manner from $\eta_D\#P = P \circ \eta_D^{-1} \in \mathcal{P}(\mathbb{R}^D)$. Similarly, for the other collection we have a sample $\{Y_i = \eta_D(\nu_i)\}_{i=1}^{N_2}$ drawn from $\eta_D\#Q$. Now, the null $H_0 : C_{\eta\#P} = C_{\eta\#Q}$ implies that the means of the distributions $\eta_D\#P$ and $\eta_D\#Q$ coincide in $\mathbb{R}^D$.

The bootstrap null distribution for $\tilde{\mathbb{T}}_{L,D'}$ can be obtained as follows. Let $\bar{X}$ and $\bar{Y}$ be the sample means; construct the combined sample $\{X_i - \bar{X} + \frac{\bar{X}+\bar{Y}}{2}\}_{i=1}^{N_1} \bigcup \{Y_i - \bar{Y} + \frac{\bar{X}+\bar{Y}}{2}\}_{i=1}^{N_2}$. This centers both samples at $\frac{\bar{X}+\bar{Y}}{2}$. Now, from the combined sample we select with replacement $N_1$ (resp. $N_2$) samples

to make bootstrap sample $\{X_i^b\}_{i=1}^{N_1}$ (resp. $\{Y_i^b\}_{i=1}^{N_2}$). Repeat this process $B$ times (we take $B = 1000$ in our experiments), and collect the null test statistic values $\tilde{\mathbb{T}}_{L,D'}^b = \tilde{\mathbb{T}}_{L,D'}(\{X_i^b\}_{i=1}^{N_1}, \{Y_i^b\}_{i=1}^{N_2})$ for $b = 1, ..., B$. The approximate $p$-value is then given by: $p = \frac{1}{B+1}\left(|\{b : \tilde{\mathbb{T}}_{L,D'}^b \geq \tilde{\mathbb{T}}_{L,D'}\}| + 1\right)$.

## 4.2  Testing via $p$-value Combination

The bootstrap test above incurs a high computational cost and the granularity of the $p$-values is determined by the number of resamples, which can be too coarse in massive multiple comparison settings often seen in industrial applications. Thus, we propose an approach that avoids resampling.

As explained above, testing $H_0 : C_{\eta\#P} = C_{\eta\#Q}$ can be interpreted as testing whether the means of the distributions $\eta_D\#P$ and $\eta_D\#Q$ coincide in $\mathbb{R}^D$. To this end, we adopt the approach proposed by [22] in a spatial statistics context. First, we apply the Behrens-Fisher-Welch $t$-test (without assuming equality of variances) to each coordinate of the samples $\{X_i = \eta_D(\mu_i)\}_{i=1}^{N_1}$ and $\{Y_i = \eta_D(\nu_i)\}_{i=1}^{N_2}$ to obtain the $p$-values $p_k, k = 1, 2, ..., D$. Second, an overall $p$-value is computed via the harmonic mean $p$-value combination method which is robust to dependencies [6, 26]: $p^H = H\left(D/(\frac{1}{p_1} + \frac{1}{p_2} + \cdots + \frac{1}{p_D})\right)$, where the function $H$ has a known form described in [26]. Another approach for combining $p$-values is the Cauchy combination test [16], but in our numerical experiments we found that the Cauchy combination approach encounters problems when any of the $p$-values is very close to 1, which can happen in our setting due to the form of the embedding $\eta_D$. Therefore, in contrast to [22], for us the harmonic combination is the only appropriate choice.

To guarantee size control, we establish a version of Theorem 1 from [16] for the harmonic mean $p$-value. Assume that a test statistic $Z \in \mathbb{R}^D$ has null distribution with zero mean and every pair of coordinates of $Z$ follows bivariate Gaussian distribution. Compute the coordinate-wise two-sided $p$-values $p_k = 2(1 - \Phi(|Z_k|))$ where $\Phi$ is the standard Gaussian CDF.

**Theorem 4.** *Let $p_k, k = 1, ..., D$ be the null $p$-values as above and $p^H$ computed via harmonic mean approach, then*

$$\lim_{\alpha \to 0} \frac{\text{Prob}\{p^H \leq \alpha\}}{\alpha} = 1.$$

In the Appendix we show that this theorem applies in our setting, so the proposed procedure asymptotically controls the size of the test for small $\alpha$. Our experimental results show that the control is already achieved for moderate sample sizes and the commonly used $\alpha = 0.05$.

# 5  Experiments

**Synthetic Experiments**  We compare the performance of our tests on distributions over finite interval, circle, and cylinder with existing methods, and settings of the embedding parameters $L, D'$. For evaluation, we use empirical power at different degrees of departure from the null hypothesis (captured by $\delta$); further details can be found in the Appendix. The summary results presented in Figure 2 show that all methods maintain nominal size (power at $\delta = 0$ is close to $\alpha$). On the inite interval: 1) from Figure 2 (a) our combination test (ISD comb) outperformed all the other tests, but the bootstrap test (ISD T boot) performs worse than others except Fmaxb; 2) from Figure 2 (b) combination test with sliced $ISW_2$ improves over the unsliced version–with more eigenfunctions, the power first improves considerably, then become similar to the unsliced version. For circular domain, Figure 2 (c) shows that our tests maintain considerably higher power than existing methods for all $\delta$. Figure 2 (d) shows that our combination test maintains nominal size on cylindrical domain.

**NHANES data on physical activity monitoring**  This data [10] contains physical activity pattern readings for 6839 individuals. Data for each individual corresponds to activity monitor intensity values for 7 days. Since the time dimension is periodic, we get person-specific probability distributions over the cylinder $S^1(T_1) \times [0, T_2)$. We check if activity patterns vary across age groups. The $p$-value combination test results are shown *below the diagonal* in Table 1. Our method detects statistically significant differences between all pairs of groups, except the 36–45 and 46–55 groups. As expected, the *control $p$-values*—obtained by mixing samples between two age groups and splitting arbitrarily—do not concentrate near zero. More details are in the Appendix.

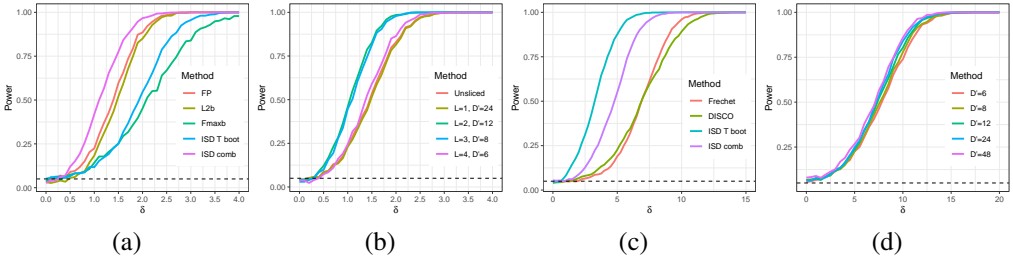

Figure 2: Performance on synthetic finite interval and manifold data. Finite interval: (a) comparison with existing methods—a test based on basis function representation (FP) [7], a sum-type $\ell_2$ norm-based test (L2b) [27], and a max-type test [28] that uses the maximum of coordinate-wise $F$ statistic (Fmaxb); (b) unsliced vs. different settings of $(L, D')$. Manifold data: (c) circular data, comparing with Fréchet ANOVA [5], and the DISCO nonparametric test [21]; (d) harmonic combination tests on cylindrical data for $L = 4$. Dotted lines indicates nominal size of all tests ($\alpha = 0.05$).

| Ages | 6–15 | 16–25 | 26–35 | 36–45 | 46–55 | 56–65 | 66–75 | 76–85 |
|---|---|---|---|---|---|---|---|---|
| 6–15 | | 0.394 | 0.098 | 0.555 | 0.882 | 0.985 | 0.919 | 0.997 |
| 16–25 | **1.2e-13** | | 0.575 | 0.967 | 0.126 | 0.921 | 0.911 | 0.977 |
| 26–35 | **3.1e-21** | **2.7e-04** | | 0.459 | 0.197 | 0.996 | 0.919 | 0.565 |
| 36–45 | **6.1e-22** | **7.9e-08** | **0.042** | | 0.864 | 0.637 | 0.849 | 0.991 |
| 46–55 | **8.2e-22** | **4.7e-05** | **0.011** | 0.343 | | 0.841 | 0.165 | 0.554 |
| 56–65 | **1.3e-25** | **0.001** | **0.001** | **5.6e-05** | **0.003** | | 0.991 | 0.962 |
| 66–75 | **3.6e-35** | **7.8e-12** | **1.5e-11** | **4.6e-15** | **1.8e-13** | **0.001** | | 0.989 |
| 76–85 | **3.8e-46** | **1.4e-26** | **1.7e-30** | **8.4e-37** | **2.1e-35** | **1.3e-17** | **6.5e-09** | |

Table 1: Activity intensity comparison across age groups in the NHANES data. Below diagonal: $p$-values for to the actual data comparisons. Above diagonal: null $p$-values obtained by combining and randomly splitting the two groups. Bold entries correspond to rejected hypotheses with the BH procedure at FDR level 0.1.

| Crime Type | Tue vs Thu | Tue vs Sat |
|---|---|---|
| Theft | 0.428 | **4.2e-06** |
| Decept Pract | 0.313 | **0.001** |
| Battery | 0.430 | **0.001** |
| Robbery | 0.119 | **0.003** |
| Narcotics | 0.854 | **0.004** |
| Criminal Dam | 0.855 | **0.02** |
| Other Offense | 0.931 | 0.052 |
| Burglary | 0.142 | 0.261 |
| Assault | 0.997 | 0.38 |
| Mot Veh Theft | 0.858 | 0.416 |

Table 2: Chicago Crime analysis $p$-values. Bold entries correspond to rejected hypotheses with the BH procedure at FDR level 0.1.

**Chicago Crime**  We use the Chicago Crimes 2018 dataset [1] to demonstrate the use of our methodology on histograms over graphs. Each beat (geographic area subdivision used by police) corresponds to a vertex, and two vertices are connected by an edge if the corresponding beats share a geographic boundary. For each crime type and day, the normalized counts of that crime type for each beat gives a daily probability distribution over the graph. Our goal is compare the collection of distributions of, say, theft occurring on Tuesday to those of Thursday and Saturday. The Tuesday versus Thursday comparison is intended as a null case, as we do not expect to see any differences between them [22]. We detect statistically significant differences between Tuesday and Saturday patterns for six categories of crime, and as expected, no differences between Tuesday and Thursday patterns. See Appendix for more details and for another graph based application in the context of Brain Connectomics.

## 6 Conclusion

The construction of $ISW_2$ provides a novel embedding of probability distributions into a Hilbert space. This can be used to adapt many inferential methods to general spaces where the existence of Fréchet means or higher moments are not guaranteed. The $ISW_2$ can also be useful for machine learning applications where prediction targets live in a general domain. Given that rigorous Fréchet mean-based methodology for such problems has only been proposed recently [9], development of prediction models for manifold-valued data that are free of restrictive assumptions is an attractive future line of research.

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
