# Intrinsic Sliced Wasserstein Distances for Comparing Collections of Probability Distributions on Manifolds and Graphs

## 1 A   Proofs and additional results

### 2 A.1   Proofs and Notes for Section 2

3 **Proposition 1.** *For $P, Q \in \mathcal{P}(\mathcal{P}(\mathcal{X}))$, the following equality holds:*

$$\mathbb{T}(P,Q) = \mathbb{E}_{\mu \sim P, \nu \sim Q}[\mathcal{D}^2(\mu,\nu)] - \frac{1}{2}\mathbb{E}_{\mu,\mu' \sim P}[\mathcal{D}^2(\mu,\mu')] - \frac{1}{2}\mathbb{E}_{\nu,\nu' \sim Q}[\mathcal{D}^2(\nu,\nu')], \tag{A.1}$$

4 *where to avoid notational clutter we use $\mathcal{D}^2(\cdot,\cdot)$ as a shorthand for $(\mathcal{D}(\cdot,\cdot))^2$.*

5 *Proof.* This is a straightforward application of the "kernel trick": using the Hilbert property of the
6 distance we can rewrite,

$$\mathbb{E}_{\mu \sim P, \nu \sim Q}[\|\eta(\mu) - \eta(\nu)\|_{\mathcal{H}}^2] - \frac{1}{2}\mathbb{E}_{\mu,\mu' \sim P}[\|\eta(\mu) - \eta(\mu')\|_{\mathcal{H}}^2] - \frac{1}{2}\mathbb{E}_{\nu,\nu' \sim Q}[\|\eta(\nu) - \eta(\nu')\|_{\mathcal{H}}^2]$$

$$= \mathbb{E}_{\mu \sim P}[\|\eta(\mu)\|_{\mathcal{H}}^2] + \mathbb{E}_{\nu \sim Q}[\|\eta(\nu)\|_{\mathcal{H}}^2] - 2\langle \mathbb{E}_{\mu \sim P}[\eta(\mu)], \mathbb{E}_{\nu \sim Q}[\eta(\nu)]\rangle_{\mathcal{H}}$$

$$- \mathbb{E}_{\mu \sim P}[\|\eta(\mu)\|_{\mathcal{H}}^2] - \mathbb{E}_{\nu \sim Q}[\|\eta(\nu)\|_{\mathcal{H}}^2]$$

$$+ \langle \mathbb{E}_{\mu \sim P}[\eta(\mu)], \mathbb{E}_{\mu \sim P}[\eta(\mu)]\rangle_{\mathcal{H}} + \langle \mathbb{E}_{\nu \sim Q}[\eta(\nu)], \mathbb{E}_{\nu \sim Q}[\eta(\nu)]\rangle_{\mathcal{H}}$$

$$= \|\mathbb{E}_{\mu \sim P}[\eta(\mu)] - \mathbb{E}_{\nu \sim Q}[\eta(\nu)]\|_{\mathcal{H}}^2 = \mathbb{T}(P,Q).$$

7 Which gives the sought equivalence. $\square$

### 8 A.2   Proofs and Notes for Section 3.1

9 **Proposition 2.** *If $\mathcal{X}$ is a smooth compact $n$-dimensional manifold and $\sum_{\ell} \lambda_{\ell}^{(n-1)/2}\alpha(\lambda_{\ell}) < \infty$,*
10 *then $ISW_2$ is well-defined.*

11 *Proof.* We use Hörmander's bound on the supremum norm of the eigenfunctions:

$$\|\phi_{\ell}\|_{\infty} \leq c\lambda_{\ell}^{(n-1)/4}\|\phi_{\ell}\|_2,$$

12 for some constant $c$ that depends on the manifold. By orthonormality of the eigenfunctions we have
13 $\forall \ell, \|\phi_{\ell}\|_2 = 1$. Next, note that $\mathcal{W}_2(\phi_{\ell}\sharp\mu, \phi_{\ell}\sharp\nu) \leq 2\|\phi_{\ell}\|_{\infty}$ as the maximum distance that the mass
14 would be transported in any transportation plan involving pushforwards via $\phi_{\ell}$ is upper bounded by
15 $2\|\phi_{\ell}\|_{\infty}$. As a result, every term in the series defining $ISW_2$ can be upper-bounded by the terms of
16 the following series:

$$\sum_{\ell} 4\|\phi_{\ell}\|_{\infty}^2 \alpha(\lambda_{\ell}) \leq \sum_{\ell} 4c^2 \lambda_{\ell}^{(n-1)/2}\alpha(\lambda_{\ell}) \propto \sum_{\ell} \lambda_{\ell}^{(n-1)/2}\alpha(\lambda_{\ell}),$$

17 which proves the claim by the direct comparison test for convergence of series. $\square$

Submitted to 36th Conference on Neural Information Processing Systems (NeurIPS 2022). Do not distribute.

*Remark* 1. When Weyl law applies, we have that $\lambda_\ell = \Theta(\ell^{2/n})$, which allows us to replace the above condition by $\sum_\ell \ell^{(n-1)/n} \alpha(\lambda_\ell) < \infty$. For the diffusion kernel/distance choice of $\alpha(\lambda) = e^{-t\lambda}$ the series always converges independently of the manifold dimension. For biharmonic choice of $\alpha(\lambda) = 1/\lambda^2$, the sufficient condition is the convergence of $\sum_\ell \ell^{(n-1)/n}/\lambda_\ell^2 \sim \sum_\ell \ell^{(n-1)/n}/(\ell^{2/n})^2 = \sum_\ell \ell^{(n-5)/n}$, where we applied Weyl's asymptotic again. As a result, the biharmonic choice of $\alpha$ is guaranteed to provide a well-defined $ISW_2$ for 1 and 2-dimensional manifolds. Notice, however, that the Hörmander's bound used in the proof of the above proposition can be rather lax in some of the settings that are practically relevant, such as the product spaces of lines and circles (where all of the eigenfunctions are bounded by a constant as can be seen from Table 1), and, thus, convergence for the biharmonic choice holds more widely.

**Proposition 3.** *If $\mathcal{D}$ is a Hilbertian probability distance such that $IS\mathcal{D}$ is well-defined, then*

*(i) $IS\mathcal{D}$ is Hilbertian, and*

*(ii) $IS\mathcal{D}$ satisfies the following metric properties: non-negativity, symmetry, the triangle inequality, and $IS\mathcal{D}(\mu, \mu) = 0$.*

*Proof.* By Hilbertian property of $\mathcal{D}$, there exists a Hilbert space $\mathcal{H}^0$ and a map $\eta^0 : \mathcal{P}(\mathbb{R}) \to \mathcal{H}^0$ such that $\mathcal{D}(\rho_1, \rho_2) = \|\eta^0(\rho_1) - \eta^0(\rho_2)\|_{\mathcal{H}^0}$ for all $\rho_1, \rho_2 \in \mathcal{P}(\mathbb{R})$. Plugging this into the definition of $IS\mathcal{D}$ we have $IS\mathcal{D}(\mu, \nu) = \|\eta(\mu) - \eta(\nu)\|_{\mathcal{H}}$, where $\mathcal{H} = \oplus_\ell \mathcal{H}^0$ and the $\ell$-th component of $\eta(\mu)$ is $\sqrt{\alpha(\lambda_\ell)} \eta_0(\phi_\ell \sharp \mu) \in \mathcal{H}^0$. The second part of Proposition 3 directly follows from the Hilbert property. $\square$

**Proposition 4.** *When $\mu = \delta_x(\cdot), \nu = \delta_y(\cdot)$ for two points $x, y \in \mathcal{X}$, we have $ISW_2(\mu, \nu) = d(x, y)$, where $d(\cdot, \cdot)$ is the spectral distance corresponding to the choice of $\alpha(\cdot)$.*

*Proof.* We have $\phi_\ell \sharp \delta_x = \delta_{\phi_\ell(x)}$ and similarly for $y$. Now $\mathcal{W}_2^2(\phi_\ell \sharp \mu, \phi_\ell \sharp \nu) = \mathcal{W}_2^2(\delta_{\phi_\ell(x)}, \delta_{\phi_\ell(y)}) = (\phi_\ell(x) - \phi_\ell(y))^2$. This last equality follows from the fact that the 2-Wasserstein on real line between delta measures is equal to the distance between the two points. Then scaling and adding up gives exactly the kernel distance $d(x, y)$ between the two points. $\square$

**Proposition 5.** *Let $\mathcal{D}(\rho_1, \rho_2) = |\mathbb{E}_{x \sim \rho_1}[x] - \mathbb{E}_{y \sim \rho_2}[y]|$ for $\rho_1, \rho_2 \in \mathcal{P}(\mathbb{R})$, then the corresponding intrinsic sliced distance is equivalent to the MMD with the spectral kernel $k(\cdot, \cdot)$.*

*Proof.* We can rewrite the definition as follows:

$$
\begin{aligned}
IS\mathcal{D}^2(\mu, \nu) &= \sum_\ell \alpha(\lambda_\ell)(\mathbb{E}_{x \sim \phi_\ell \sharp \mu}[x] - \mathbb{E}_{y \sim \phi_\ell \sharp \nu}[y])^2 = \sum_\ell \alpha(\lambda_\ell)(\mathbb{E}_{x \sim \mu}[\phi_\ell(x)] - \mathbb{E}_{y \sim \nu}[\phi_\ell(y)])^2 \\
&= \sum_\ell \alpha(\lambda_\ell)(\mathbb{E}_{x,x' \sim \mu}[\phi_\ell(x)\phi_\ell(x')] + \mathbb{E}_{y,y' \sim \nu}[\phi_\ell(y)\phi_\ell(y')] - 2\mathbb{E}_{x \sim \mu, y \sim \nu}[\phi_\ell(x)\phi_\ell(y)]) \\
&= \mathbb{E}_{x,x' \sim \mu}[\sum_\ell \alpha(\lambda_\ell)\phi_\ell(x)\phi_\ell(x')] + \mathbb{E}_{y,y' \sim \nu}[\sum_\ell \alpha(\lambda_\ell)\phi_\ell(y)\phi_\ell(y')] \\
&\quad - 2\mathbb{E}_{x \sim \mu, y \sim \nu}[\sum_\ell \alpha(\lambda_\ell)\phi_\ell(x)\phi_\ell(y)] \\
&= \mathbb{E}_{x,x' \sim \mu}[k(x, x')] + \mathbb{E}_{y,y' \sim \nu}[k(y, y')] - 2\mathbb{E}_{x \sim \mu, y \sim \nu}[k(x, y)],
\end{aligned}
$$

where we used the spectral kernel $k(x, y) = \sum_\ell \alpha(\lambda_\ell)\phi_\ell(x)\phi_\ell(y)$. The last expression coincides with the MMD based on kernel $k(\cdot, \cdot)$; see Lemma 6 in [12]. $\square$

**Proposition 6.** $MMD(\mu, \nu) \leq ISW_2(\mu, \nu)$ *when the same $\alpha(\cdot)$ is used in both constructions.*

*Proof.* This follows directly from the fact that for $\rho_1, \rho_2 \in \mathcal{P}(\mathbb{R})$ the inequality $|\mathbb{E}_{x \sim \rho_1}[x] - \mathbb{E}_{y \sim \rho_2}[y]| \leq \mathcal{W}_1(\rho_1, \rho_2) \leq \mathcal{W}_2(\rho_1, \rho_2)$ holds. Here the first inequality follows from the centroid bound [22], and the second inequality is the well-known ordering property of Wasserstein distances [25]. $\square$

**Theorem 1.** *If $\alpha(\lambda) > 0$ for all $\lambda > 0$, then $ISW_2$ is a metric on $\mathcal{P}(\mathcal{X})$.*

54 *Proof.* In the light of the Proposition 3 it remains only to prove that $ISW_2(\mu, \nu) = 0$ implies
55 $\mu = \nu$. According to Proposition 6, $ISW_2(\mu, \nu) = 0$ yields $MMD(\mu, \nu) = 0$. The assumption that
56 $\alpha(\lambda) > 0$ for all $\lambda > 0$ implies that the spectral kernel $k(\cdot, \cdot)$ corresponding to $\alpha(\cdot)$ is universal [17].
57 Universality implies the characteristic property [12], which in turn means that $MMD(\mu, \nu) = 0$ is
58 equivalent to $\mu = \nu$, proving the claim. $\square$

59 **Proposition 7.** *There exists a constant $c$ depending only on $\mathcal{X}$ such that for all $\mu, \nu \in \mathcal{P}(\mathcal{X})$ the*
60 *inequality $ISW_2(\mu, \nu) \le c\mathcal{W}_2^{\mathcal{X}}(\mu, \nu)\sqrt{\sum_\ell \lambda_\ell^{(n+3)/2}\alpha(\lambda_\ell)}$ holds; here, $n$ is the dimension of $\mathcal{X}$.*

61 *Proof.* We remind $\mathcal{W}_2^{\mathcal{X}}$ is the 2-Wasserstein distance defined directly $\mathcal{P}(\mathcal{X})$ using the geodesic
62 distance as the ground metric. The Neumann eigenfunctions on compact manifolds satisfy the
63 inequality $\|\nabla\phi_\ell\|_\infty \le c_1\lambda_\ell\|\phi_\ell\|_\infty$, see [13]. Applying the bound used in the proof of convergence,
64 $\|\phi_\ell\|_\infty \le c_2\lambda_\ell^{(n-1)/4}$, we get that $\phi_\ell$ is Lipschitz with respect to the geodesic distance on $\mathcal{X}$ with the
65 Lipschitz constant bounded by $c\lambda_\ell\lambda_\ell^{(n-1)/4} = c\lambda_\ell^{(n+3)/4}$.

66 Consider the optimal coupling between $\mu$ and $\nu$ whose cost equals to $\mathcal{W}_2^{\mathcal{X}}(\mu, \nu)$. Note that this
67 coupling straightforwardly provides a coupling between the pushforwards $\phi_\ell\sharp\mu$ and $\phi_\ell\sharp\nu$. Using the
68 Lipschitz property of eigenfunctions, we see that the cost of the pushforward coupling is smaller
69 than $c\lambda_\ell^{(n+3)/4}\mathcal{W}_2^{\mathcal{X}}(\mu, \nu)$. Since any such coupling provides an upper bound on $\mathcal{W}_2(\phi_\ell\sharp\mu, \phi_\ell\sharp\nu)$, we
70 have $\mathcal{W}_2(\phi_\ell\sharp\mu, \phi_\ell\sharp\nu) \le c\lambda_\ell^{(n+3)/4}\mathcal{W}_2^{\mathcal{X}}(\mu, \nu)$. Plugging this into the formula for $ISW_2$ we get the
71 claimed bound. $\square$

72 **Proposition 8.** *Let $\{\mu_i\}_{i=1}^N$ and $\{\nu_i\}_{i=1}^N$ be two collections of probability measures on $\mathcal{P}(\mathcal{X})$, such*
73 *that $\forall i, \mathcal{W}_2^{\mathcal{X}}(\mu_i, \nu_i) \le \epsilon$, then $\mathbb{T}(\{\mu_i\}_{i=1}^N, \{\nu_i\}_{i=1}^N) \le C^2\epsilon^2$. Here $C = c\sqrt{\sum_\ell \lambda_\ell^{(n+3)/2}\alpha(\lambda_\ell)}$ from*
74 *previous proposition and is assumed to be finite.*

75 *Proof.* We have

$$
\begin{aligned}
\mathbb{T}(\{\mu_i\}_{i=1}^N, \{\nu_i\}_{i=1}^N) &= \left\|\frac{1}{N}\sum_{i=1}^N \eta(\mu_i) - \frac{1}{N}\sum_{i=1}^N \eta(\nu_i)\right\|_{\mathcal{H}}^2 = \left\|\frac{1}{N}\sum_{i=1}^N (\eta(\mu_i) - \eta(\nu_i))\right\|_{\mathcal{H}}^2 \\
&\le \frac{1}{N}\sum_{i=1}^N \|\eta(\mu_i) - \eta(\nu_i)\|_{\mathcal{H}}^2 = \frac{1}{N}\sum_{i=1}^N ISW_2^2(\mu_i, \nu_i) \le \frac{1}{N}\sum_{i=1}^N (C\mathcal{W}_2^{\mathcal{X}}(\mu_i, \nu_i))^2 \\
&\le \frac{1}{N}N(C\epsilon)^2 = C^2\epsilon^2.
\end{aligned}
$$

76 $\square$

## A.3 Computational Details for Section 3.2

78 The case of finite intervals is the building block for the general case, so let us first consider the
79 case of $\mathcal{X} = [0, T]$. We represent a histogram over this interval by a discrete measure of the form
80 $\mu = \sum w_a\delta_{x_a}$ with the histogram bin centers $x_a \in [0, T]$ and weights $w_a$ satisfying $\sum w_a = 1$,
81 where $a = 1, 2, ..., A$. Note that it is not required for the histograms in the collections to be supported
82 at the same bin locations. For a given histogram, let $\{x_{(a)}, w_{(a)}\}_{a=1}^A$ be the locations sorted from
83 smallest to largest and their corresponding weights; since the bin locations are unique there will not
84 be any ties. The quantile function is computed via $F_\mu^{-1}(s) := \min\{x_{(a)} : \sum_{b \le a} w_{(b)} > s\}$. The
85 approximate map $\eta_{D'}^0$ now can be computed using the $s_k$-th quantile value $F_\mu^{-1}(s_k)$ for each value of
86 $s_k, k = 1, ..., D'$.

87 For a general domain $\mathcal{X}$, the histogram representation is the same as above: $\sum w_a\delta_{x_a}$ with the
88 histogram bin centers $x_a \in \mathcal{X}$ and weights $w_a$ satisfying $\sum w_a = 1$, where $a = 1, 2, ..., A$. The
89 pushforward $\phi_\ell\sharp\mu$ gives a histogram on the real line defined by $\sum w_a\delta_{\phi_\ell(x_a)}$. Note that while $x_a$
90 are distinct, their images under $\phi_\ell$ do not have to be distinct, so one re-aggregates the weights to
91 obtain $\sum_{a \in S} w_a'\delta_{\phi_\ell(x_a)}$, where $S$ is a subset of $1, 2, ..., A$ and $w_a'$ are the new weights. It is now
92 straightforward to compute the quantile function as before and build the approximate map $(\eta_D)_\ell$.
93 Doing so for the different values of $\ell$ and concatenating the resulting vectors gives $\eta_D$.

| $\mathcal{X}$ | Eigenvalues | Eigenfunctions |
|---|---|---|
| $[0, T]$ | $(\frac{\pi\ell}{T})^2$ | $\sqrt{\frac{2}{T}}\cos\frac{\pi\ell x}{T}$ |
| $S^1(T) = [0, T] \mod T$ | $(\frac{2\pi\ell}{T})^2$ | $\sqrt{\frac{2}{T}}[\cos/\sin]\frac{2\pi\ell x}{T}$ |
| $[0, T_1] \times [0, T_2]$ | $(\frac{\pi\ell_1}{T_1})^2 + (\frac{\pi\ell_2}{T_2})^2$ | $\sqrt{\frac{4}{T_1 T_2}}\cos\frac{\pi\ell_1 x}{T_1}\cos\frac{\pi\ell_2 x}{T_2}$ |
| $S^1(T_1) \times [0, T_2]$ | $(\frac{2\pi\ell_1}{T_1})^2 + (\frac{\pi\ell_2}{T_2})^2$ | $\sqrt{\frac{4}{T_1 T_2}}[\cos/\sin]\frac{2\pi\ell_1 x}{T_1}\cos\frac{\pi\ell_2 x}{T_2}$ |
| $S^1(T_1) \times S^1(T_2)$ | $(\frac{2\pi\ell_1}{T_1})^2 + (\frac{2\pi\ell_2}{T_2})^2$ | $\sqrt{\frac{4}{T_1 T_2}}[\cos/\sin]\frac{2\pi\ell_1 x}{T_1}[\cos/\sin]\frac{\pi\ell_2 x}{T_2}$ |
| $S^2$ | | Spherical harmonics [5] |
| Graphs/Data Clouds/Meshes | | Eigen-decomposition of the Laplacian matrix |

Table 1: Eigenvalues and eigenfunctions of the Laplace-Beltrami operator with Neumann boundary conditions for simple manifolds. We exclude zero eigenvalue and the corresponding constant eigenvector; thus, all indices $\ell, \ell_1, \ell_2$ run over positive integers. The notation $[\cos/\sin]$ means picking either the cosine or sine function—*all choices must be used, giving multiple eigenfunctions.*

In practice, these computations can be carried out on a variety of domains—analytic manifolds, manifolds discretized as point clouds or meshes, and graphs. In most cases the spectral decomposition of the Laplace-Beltrami operator or graph Laplacian has to be computed numerically [7, 20]. For applications that involve simple manifolds, the eigenvalues and eigenfunctions can be computed analytically. For completeness we list them in Table 1. Note that we benefit from the fact that the eigen-decomposition for product spaces can be derived from the eigen-decompositions of the components.

The choice of the function $\alpha(\cdot)$ determining the contributions of each spectral band is problem specific. When working on manifolds of low dimension, the choice of $\alpha(\cdot)$ that corresponds to the biharmonic distance is convenient. While the diffusion distance provides a general choice that works on manifolds of any dimension, the biharmonic distance does not have any parameters to tune and was shown to provide an excellent alternative to the geodesic distance in low-dimensional settings [14]. When in doubt, inspecting the behavior of the distance on the underlying domain will allow assessing whether the distance is appropriate for the given problem.The importance of relying on a well-behaved spectral distance was highlighted in Proposition 4.

### A.4 Proofs and Notes for Section 4.1

We remind that we will be using the following test statistic for the results that are discussed below:

$$\hat{\mathbb{T}} \equiv \sum_{i,j} \frac{I\mathcal{SW}_2^2(\mu_i, \nu_j)}{N_1 N_2} - \sum_{i,j:i\neq j} \frac{I\mathcal{SW}_2^2(\mu_i, \mu_j)}{2N_1(N_1-1)} - \sum_{i,j:i\neq j} \frac{I\mathcal{SW}_2^2(\nu_i, \nu_j)}{2N_2(N_2-1)}. \tag{A.2}$$

**Proposition 9.** *Assume conditions (i)-(iii) hold. Define $N = N_1 + N_2$, and assume that as $N_1, N_2 \to \infty$, we have $N_1/N \to \rho_1$, $N_2/N \to \rho_2 = 1 - \rho_1$, for some fixed $0 < \rho_1 < 1$. Define a new measure $R$ as a scaled mixture of the centered pushforward measures*

$$R = \left(\frac{1}{\rho_1} + \frac{1}{\rho_2}\right)^{-1}\left[\frac{1}{\rho_1}(\eta\#P - C_{\eta\#P}) + \frac{1}{\rho_2}(\eta\#Q - C_{\eta\#Q})\right] = \rho_2(\eta\#P - C_{\eta\#P}) + \rho_1(\eta\#Q - C_{\eta\#Q}).$$

*Suppose $\gamma_m, m = 1, 2, \ldots$ are the eigenvalues of*

$$\frac{1}{\rho_1\rho_2}\int_{\mathcal{H}}\langle x, x'\rangle_{\mathcal{H}}\psi_m(x')dR(x') = \gamma_m\psi_m(x).$$

*Then under $H_0 : C_{\eta\#P} = C_{\eta\#Q}$ we have*

$$N\hat{\mathbb{T}} \rightsquigarrow \sum_{m=1}^{\infty}\gamma_m(A_m^2 - 1), \tag{A.3}$$

where $A_m$ are i.i.d. $\mathcal{N}(0,1)$ random variables. Under $H_1 : C_{\eta\#P} \neq C_{\eta\#Q}$ we have $\sqrt{N}(\hat{\mathbb{T}} - \mathbb{T}) \rightsquigarrow N(0, \sigma_1^2)$, where

$$
\sigma_1^2 = 4\left[\frac{1}{\rho_1}\mathbb{V}_{\mu\sim P}\mathbb{E}_{\mu'\sim P}\langle\eta(\mu), \eta(\mu')\rangle_{\mathcal{H}} + \frac{1}{\rho_2}\mathbb{V}_{\nu\sim Q}\mathbb{E}_{\nu'\sim Q}\langle\eta(\nu), \eta(\nu')\rangle_{\mathcal{H}} + \right.
$$
$$
\left. \frac{1}{\rho_1}\mathbb{V}_{\mu\sim P}\mathbb{E}_{\nu\sim Q}\langle\eta(\mu), \eta(\nu)\rangle_{\mathcal{H}} + \frac{1}{\rho_2}\mathbb{V}_{\nu\sim Q}\mathbb{E}_{\mu\sim P}\langle\eta(\mu), \eta(\nu)\rangle_{\mathcal{H}}\right]. \tag{A.4}
$$

*Proof.* Using the Hilbertianity of $ISD$ (Proposition 3), we have

$$
\begin{aligned}
ISD^2(\mu_i, \mu_j) &= \|\eta(\mu_i) - \eta(\mu_j)\|_{\mathcal{H}}^2 \\
&= \|\eta(\mu_i)\|_{\mathcal{H}}^2 + \|\eta(\mu_j)\|_{\mathcal{H}}^2 - 2\langle\eta(\mu_i), \eta(\mu_j)\rangle_{\mathcal{H}},
\end{aligned}
$$

Consequently

$$
\sum_{i,j:i\neq j} ISD^2(\mu_i, \mu_j) = 2(N_1 - 1)\sum_{i=1}^{N_1}\|\eta(\mu_i)\|_{\mathcal{H}}^2 - 2\sum_{i,j:i\neq j}\langle\eta(\mu_i), \eta(\mu_j)\rangle_{\mathcal{H}}.
$$

Similarly,

$$
\sum_{i,j:i\neq j} ISD^2(\nu_i, \nu_j) = 2(N_2 - 1)\sum_{i=1}^{N_2}\|\eta(\nu_i)\|_{\mathcal{H}}^2 - 2\sum_{i,j:i\neq j}\langle\eta(\nu_i), \eta(\nu_j)\rangle_{\mathcal{H}},
$$
$$
\sum_{i,j} ISD^2(\mu_i, \nu_j) = N_2\sum_{i=1}^{N_1}\|\eta(\mu_i)\|_{\mathcal{H}}^2 + N_1\sum_{j=1}^{N_2}\|\eta(\nu_j)\|_{\mathcal{H}}^2 - 2\sum_{i,j:i\neq j}\langle\eta(\mu_i), \eta(\nu_j)\rangle_{\mathcal{H}}.
$$

Putting these back into Eq. (A.2) after simplifying and cancelling out the norm-square terms we have

$$
\hat{\mathbb{T}} = \frac{1}{N_1(N_1 - 1)}\sum_{i,j:i\neq j}\langle\eta(\mu_i), \eta(\mu_j)\rangle_{\mathcal{H}} + \frac{1}{N_2(N_2 - 1)}\sum_{i,j:i\neq j}\langle\eta(\nu_i), \eta(\nu_j)\rangle_{\mathcal{H}}
$$
$$
- \frac{2}{N_1 N_2}\sum_{i,j}\langle\eta(\mu_i), \eta(\nu_j)\rangle_{\mathcal{H}}. \tag{A.5}
$$

At this point, we replace the maps $\eta$ by their centered versions $\tilde{\eta}(\mu) = \eta(\mu) - C_{\eta\#P}, \tilde{\eta}(\nu) = \eta(\nu) - C_{\eta\#Q}$; remember that the center of mass of $\eta\#P$ is denoted by $C_{\eta\#P}$. Accumulating the sample-level partial sums above the centering terms cancel out under $H_0 : C_{\eta\#P} = C_{\eta\#Q}$, so that each $\eta$ can be replaced by $\tilde{\eta}$ in (A.5) above.

Denote $x_i \equiv \tilde{\eta}(\mu_i), y_i \equiv \tilde{\eta}(\nu_i)$ as the Hilbert-embedded samples of $X \sim \tilde{\eta}\#P, Y \sim \tilde{\eta}\#Q$, respectively. We remind now that $R$ is a mixture of the centered pushforward measures: $R = \rho_2(\tilde{\eta}\#P) + \rho_1(\tilde{\eta}\#Q)$. Let $L_2(\mathcal{H}, R)$ be the space of real-valued functions on $\mathcal{H}$ that are square integrable with respect to $R$. Now we can define the following operator $S : L_2(\mathcal{H}, R) \to \mathcal{H}$,

$$
(Sf)(x) := \int_{\mathcal{H}}\langle x, x'\rangle_{\mathcal{H}} f(x')dR(x').
$$

Following condition (ii), $\langle\cdot, \cdot\rangle_{\mathcal{H}}$ is square-integrable under $R$. The above operator is thus Hilbert-Schmidt, hence compact [19, Theorem VI.23]. Consequently, it permits an eigenfunction decomposition with respect to measure $R$, $\langle x, x'\rangle_{\mathcal{H}} = \sum_{m=1}^{\infty}\gamma_m\psi_m(x)\psi_m(x')$, for $x, x' \in \mathcal{H}$. Note that here $\psi_m : \mathcal{H} \to \mathbb{R}$ and

$$
\int_{\mathcal{H}}\langle x, x'\rangle\psi_m(x')dR(x') = \gamma_m\psi_m(x),
$$

$$
\int_{\mathcal{H}}\psi_m(x)\psi_n(x)dR(x) = \delta_{mn}.
$$

Due to the centering of $\eta$ we also have when $\gamma_m \neq 0$,

$$
\gamma_m\mathbb{E}_X[\psi_m(x)] = \int_{\mathcal{H}}\mathbb{E}_X[\langle x, x'\rangle_{\mathcal{H}}]\psi_n(x')dR(x') = 0 \quad \Rightarrow \quad \mathbb{E}_X[\psi_m(x)] = 0.
$$

Similarly, $\mathbb{E}_Y[\psi_m(y)] = 0$. The V-statistic from the overall sample can now be written as an infinite sum [24, Section 5.5]:

$$\|\hat{C}_{\eta\#P} - \hat{C}_{\eta\#Q}\|_{\mathcal{H}}^2 = \sum_{m=1}^{\infty} \gamma_m \left( \frac{1}{N_1} \sum_{i=1}^{N_1} \psi_m(x_i) - \frac{1}{N_2} \sum_{i=1}^{N_2} \psi_m(y_i) \right)^2 := \sum_{m=1}^{\infty} \gamma_m a_m^2.$$

Our goal is to show that (a) $a_m \rightsquigarrow \mathcal{N}(0, (N\rho_1\rho_2)^{-1})$, for $\forall m$, and (b) $a_m$ and $a_n$ are independent when $m \neq n$.

First note that

$$\mathbb{E}(a_m) = \mathbb{E} \left( \frac{1}{N_1} \sum_{i=1}^{N_1} \psi_m(x_i) - \frac{1}{N_2} \sum_{i=1}^{N_2} \psi_m(y_i) \right) = 0.$$

In addition we have,

$$
\begin{aligned}
Cov(a_m, a_n) &= \mathbb{E}(a_m a_n) - \mathbb{E}(a_m).\mathbb{E}(a_n) \\
&= \mathbb{E}(a_m a_n) \\
&= \mathbb{E} \left( \frac{1}{N_1} \sum_{i=1}^{N_1} \psi_m(x_i) - \frac{1}{N_2} \sum_{i=1}^{N_2} \psi_m(y_i) \right) \left( \frac{1}{N_1} \sum_{i=1}^{N_1} \psi_n(x_i) - \frac{1}{N_2} \sum_{i=1}^{N_2} \psi_n(y_i) \right) \\
&= \mathbb{E}_X \left( \frac{1}{N_1^2} \sum_{i=1}^{N_1} \psi_m(x_i)\psi_n(x_i) \right) + \mathbb{E}_Y \left( \frac{1}{N_2^2} \sum_{i=1}^{N_2} \psi_m(y_i)\psi_n(y_i) \right) \\
&= \frac{1}{\rho_1 N} \mathbb{E}_X \left( \frac{1}{N_1} \sum_{i=1}^{N_1} \psi_m(x_i)\psi_n(x_i) \right) + \frac{1}{\rho_2 N} \mathbb{E}_Y \left( \frac{1}{N_2} \sum_{i=1}^{N_2} \psi_m(y_i)\psi_n(y_i) \right) \\
&= \frac{1}{N} \left[ \frac{1}{\rho_1} \int_{\mathcal{H}} \psi_m(x)\psi_n(x)d(\tilde{\eta}\#P)(x) + \frac{1}{\rho_2} \int_{\mathcal{H}} \psi_m(y)\psi_n(y)d(\tilde{\eta}\#Q)(y) \right] \\
&= \frac{1}{N\rho_1\rho_2} \int_{\mathcal{H}} \psi_m(z)\psi_n(z)dR(z) \\
&= \frac{1}{N\rho_1\rho_2} \delta_{mn}.
\end{aligned}
$$

An application of CLT follows that (a) holds. This together with vanishing covariance proves (b). Consequently, we can apply the CLT for degenerate V-statistics [24, Section 5.5.2] to obtain the limiting distribution, with $A_m \sim \mathcal{N}(0, 1)$,

$$N\|\hat{C}_{\eta\#P} - \hat{C}_{\eta\#Q}\|_{\mathcal{H}}^2 \rightsquigarrow \sum_{m=1}^{\infty} \frac{\gamma_m}{\rho_1\rho_2} A_m^2.$$

Let us now look at the difference between this V-statistic and our U-statistic, i.e. $\hat{\mathbb{T}}$ in (A.5). We see that

$$
\begin{aligned}
\|\hat{C}_{\eta\#P} - \hat{C}_{\eta\#Q}\|_{\mathcal{H}}^2 - \hat{\mathbb{T}} &= \frac{1}{N_1^2} \sum_{i,j} \langle x_i, x_j \rangle_{\mathcal{H}} + \frac{1}{N_2^2} \sum_{i,j} \langle y_i, y_j \rangle_{\mathcal{H}} - \frac{2}{N_1 N_2} \sum_{i,j} \langle x_i, y_j \rangle_{\mathcal{H}} \\
&\quad - \frac{1}{N_1(N_1-1)} \sum_{i,j;i\neq j} \langle x_i, x_j \rangle_{\mathcal{H}} + \frac{1}{N_2(N_2-1)} \sum_{i,j;i\neq j} \langle y_i, y_j \rangle_{\mathcal{H}} + \frac{2}{N_1 N_2} \sum_{i,j} \langle x_i, y_j \rangle_{\mathcal{H}} \\
&= - \left[ \frac{1}{N_1(N_1-1)} - \frac{1}{N_1^2} \right] \sum_{i,j;i\neq j} \langle x_i, x_j \rangle_{\mathcal{H}} - \left[ \frac{1}{N_2(N_2-1)} - \frac{1}{N_2^2} \right] \sum_{i,j;i\neq j} \langle y_i, y_j \rangle_{\mathcal{H}} \\
&\quad + \left( \frac{1}{N_1^2} \sum_{i=1}^{N_1} \|x_i\|_{\mathcal{H}}^2 + \frac{1}{N_2^2} \sum_{i=1}^{N_2} \|y_i\|_{\mathcal{H}}^2 \right) \\
&= -K^x - K^y + B.
\end{aligned}
$$

146 We claim that $K^x = O_p(N_1^{-2})$, $K^y = O_p(N_2^{-2})$, and $NB \xrightarrow{P} \sum_{m=1}^{\infty} \gamma_m(\rho_1\rho_2)^{-1}$. As a result,

$$N\left[\|\hat{C}_{\eta\#P} - \hat{C}_{\eta\#Q}\|_{\mathcal{H}}^2 - \hat{\mathbb{T}}\right] = -NO_p(N_1^{-2}) - NO_p(N_2^{-2}) + \sum_{m=1}^{\infty} \frac{\gamma_m}{\rho_1\rho_2} + o_p(1)$$

$$= \sum_{m=1}^{\infty} \frac{\gamma_m}{\rho_1\rho_2} + o_p(1),$$

147 so that $N\hat{\mathbb{T}} \rightsquigarrow \sum_{m=1}^{\infty} \gamma_m(\rho_1\rho_2)^{-1}(A_m^2 - 1)$, and we conclude the proof by reassigning $\gamma_m \leftarrow$
148 $\gamma_m(\rho_1\rho_2)^{-1}$ to obtain (A.3).

149 **Proof of Claim.** For the $K$-terms we have

$$K^x = \left[\frac{1}{N_1(N_1-1)} - \frac{1}{N_1^2}\right] \sum_{i,j;i\neq j} \langle x_i, x_j\rangle_{\mathcal{H}}$$

$$= \frac{1}{N_1^2(N_1-1)} \sum_{i,j;i\neq j} \langle x_i, x_j\rangle_{\mathcal{H}}$$

$$= \sum_{m=1}^{\infty} \gamma_m \frac{1}{N_1} \frac{1}{N_1(N_1-1)} \sum_{i,j;i\neq j} \psi_m(x_i)\psi_m(x_j)$$

$$= \sum_{m=1}^{\infty} \gamma_m K_m^x,$$

150 where $K_m^x$ is defined as the inner sum. Since $\mathbb{E}_X\psi_m(x) = 0$, we have $\mathbb{E}_X(K_m^x) =$
151 $\frac{1}{N_1}[\mathbb{E}_X\psi_m(x)]^2 = 0$, and

$$Var_X(K_m^x) = \mathbb{E}_X[(K_m^x)^2]$$

$$= \frac{1}{N_1^2}\mathbb{E}_X\left[\frac{1}{N_1^2(N_1-1)^2} \sum_{i\neq j}\sum_{l\neq k} \psi_m(x_i)\psi_m(x_j)\psi_m(x_l)\psi_m(x_k)\right] \quad \text{(A.6)}$$

$$= \frac{1}{N_1^2}\mathbb{E}_X\left[\frac{1}{N_1^2(N_1-1)^2} \sum_{i\neq j} \psi_m^2(x_i)\psi_m^2(x_j)\right] \quad \text{(A.7)}$$

$$= \frac{1}{N_1^2} \cdot \frac{1}{N_1(N_1-1)} \left(\mathbb{E}_X[\psi_m^2(x)]\right)^2.$$

152 The cross terms—terms involving $l \neq i$ or $k \neq j$—vanish due to the sample being iid and eigenfunc-
153 tions having zero expectations. The expectation in the last line is finite by assumption (ii), so that
154 $Var_X(K_m^x) = O(N_1^{-4})$, giving $K_m^x = O_p(N_1^{-2})$. Note that the assumption (ii) moreover implies
155 the convergence of the big-oh coefficients, leading to $K^x = \sum_{m=1}^{\infty} \gamma_m K_m^x = O_p(N_1^{-2})$. Similarly
156 we get $K^y = O_p(N_2^{-2})$.

157 For the term $B$, we have

$$B = \frac{1}{N_1^2}\sum_{i=1}^{N_1}\|x_i\|_{\mathcal{H}}^2 + + \frac{1}{N_2^2}\sum_{i=1}^{N_2}\|y_i\|_{\mathcal{H}}^2 = \sum_{m=1}^{\infty}\gamma_m\left[\frac{1}{N_1^2}\sum_{i=1}^{N_1}\psi_m^2(x_i) + \frac{1}{N_2^2}\sum_{i=1}^{N_2}\psi_m^2(y_i)\right] := \sum_{m=1}^{\infty}\gamma_m C_m.$$

158 Taking expectation,

$$\mathbb{E}_{X,Y}(C_m) = \frac{1}{\rho_1 N}\int_{\mathcal{H}}\psi_m^2(x)d(\tilde{\eta}\#P)(x) + \frac{1}{\rho_2 N}\int_{\mathcal{H}}\psi_m^2(y)d(\tilde{\eta}\#Q)(y)$$

$$= \frac{1}{N\rho_1\rho_2}\int_{\mathcal{H}}\psi_m^2(z)dR(z)$$

$$= \frac{1}{N\rho_1\rho_2}.$$

159     Thus $\mathbb{E}_{X,Y}(NB) = \sum_m \gamma_m (\rho_1 \rho_2)^{-1}$. Finally,

$$NB = \sum_{m=1}^{\infty} \gamma_m \left[ \frac{1}{\rho_1 N_1} \sum_{i=1}^{N_1} \psi_m^2(x_i) + \frac{1}{\rho_2 N_2} \sum_{i=1}^{N_2} \psi_m^2(y_i) \right] \xrightarrow{P} \sum_{m=1}^{\infty} \gamma_m \left[ \frac{1}{\rho_1} \mathbb{E}_X \psi_m^2(x) + \frac{1}{\rho_2} \mathbb{E}_Y \psi_m^2(y) \right] = \mathbb{E}_{X,Y}(NB)$$

160     by the weak law of large numbers. This proves the claim for $B$.

161     **Alternative Distribution.** For the the limiting distribution under $H_1$, notice that the first two terms
162     in (A.2) are the one-sample U-statistic calculated on the samples $\{\mu_i\}_{i=1}^{N_1}$ and $\{\nu_i\}_{i=1}^{N_2}$, respectively.
163     Using the CLT for non-degenerate U-statistics [24, Section 5.5.1, Theorem A], we have

$$\sqrt{N_1} \left[ \frac{\sum_{i,j:i\neq j}\langle \eta(\mu_i), \eta(\mu_j)\rangle_{\mathcal{H}}}{N_1(N_1-1)} - \mathbb{E}_{\mu,\mu'\sim P}\langle \eta(\mu), \eta(\mu')\rangle_{\mathcal{H}} \right] \rightsquigarrow N\left(0, 4\mathbb{V}_{\mu\sim P}\left[\mathbb{E}_{\mu'\sim P}\langle \eta(\mu),\eta(\mu')\rangle_{\mathcal{H}}\right]\right),$$

$$\sqrt{N_2} \left[ \frac{\sum_{i,j:i\neq j}\langle \eta(\nu_i), \eta(\nu_j)\rangle_{\mathcal{H}}}{N_2(N_2-1)} - \mathbb{E}_{\nu,\nu'\sim Q}\langle \eta(\nu), \eta(\nu')\rangle_{\mathcal{H}} \right] \rightsquigarrow N\left(0, 4\mathbb{V}_{\nu\sim Q}\left[\mathbb{E}_{\nu'\sim Q}\langle \eta(\nu),\eta(\nu')\rangle_{\mathcal{H}}\right]\right).$$

164     For the third summand, using an equivalent CLT for two-sample U-statistic [8, Theorem 2.1],

$$\sqrt{N}\left[ \frac{\sum_{i,j}\langle \eta(\mu_i), \eta(\nu_j)\rangle_{\mathcal{H}}}{N_1 N_2} - \mathbb{E}_{\mu\sim P, \nu\sim Q}\langle \eta(\mu), \eta(\nu)\rangle_{\mathcal{H}} \right] \rightsquigarrow$$

$$N\left(0, \frac{1}{\rho_1}\mathbb{V}_{\mu\in P}\left[\mathbb{E}_{\nu\sim Q}\langle \eta(\mu),\eta(\nu)\rangle_{\mathcal{H}}\right] + \frac{1}{\rho_2}\mathbb{V}_{\nu\in Q}\left[\mathbb{E}_{\mu\sim P}\langle \eta(\mu),\eta(\nu)\rangle_{\mathcal{H}}\right]\right).$$

165     We obtain (A.4) by combining the above three results. $\qquad\square$

166     The following result now ensures that approximations of $\hat{\mathbb{T}}$ using the top few eigenfunctions and a
167     finite number of CDF embeddings can be constructed with small approximation errors, provided the
168     manifold eigenvalues are declining suitably fast and the finite dimensional $\eta_D(\cdot)$ is suitably smooth.

169     **Proposition 10.** *Suppose that (i), (ii) and (iii) hold. Then we have* $\sqrt{N}(\hat{\mathbb{T}} - \hat{\mathbb{T}}_{L_N}) = o_p(1)$ *and*
170     $\sqrt{N}(\hat{\mathbb{T}}_{L_N} - \tilde{\mathbb{T}}_{L_N,D_N}) = o_p(1)$ *for the following choices of* $L_N, D_N$:

$$L_N \geq \min_{L'}\left\{ L' : \sum_{\ell=L'+1}^{\infty} \alpha_\ell \lambda_\ell^{(n+3)/2} \leq \frac{1}{N^{1+\delta}} \right\}, \quad D_N \geq kc^2 N^{1+\delta} \sum_{l=1}^{L_N} \alpha_\ell \lambda_\ell^{(n-1)/2},$$

171     *where* $\delta, k > 0$ *are constants depending only on* $\mathcal{X}$.

172     As we mention in the discussion after condition (i), for the heat kernel with tuning parameter $t$:
173     $\alpha(\lambda) = \exp(-t\lambda)$, the assumption (i) that $\sum_{\ell=1}^{\infty} \alpha_l \lambda_\ell^{(n+3)/2} < \infty$ holds. The bound on $D_N$ is a
174     consequence of classical bounds on Riemann approximation errors in terms of $\|\eta'\|_\infty$. Absolute
175     continuity of $\mu \sim P, \nu \sim Q$ ensures the existence of $(F_{\phi_\ell \sharp \mu}^{-1})'(s), (F_{\phi_\ell \sharp \nu}^{-1})'(s)$ (where prime denotes
176     the derivative) for Lebesgue-almost every $s \in [0,1]$ [10, Lemma 2.3].

177     *Proof.* Notice that given $L_N$, summands in the expression $\hat{\mathbb{T}} - \hat{\mathbb{T}}_{L_N}$ are the tail sums
178     $\sum_{\ell=L_N+1}^{\infty} \alpha_l \mathcal{W}_2^2(\phi_\ell \sharp \cdot, \phi_\ell \sharp \cdot)$ starting at the $L_N + 1^{\text{th}}$ term. Using a similar approach as the proof
179     of Proposition 7, this is bounded above by a scalar multiple of the geodesic distance, specifically
180     $c\mathcal{W}_2^{\mathcal{X}}(\cdot, \cdot)\sqrt{\sum_{\ell=L_N+1}^{\infty} \alpha_\ell \lambda_\ell^{(n+3)/2}}$. By assumption $\sum_{\ell=1}^{\infty} \alpha_\ell \lambda_\ell^{(n+3)/2} < \infty$, so that given $\epsilon > 0$ we
181     can always choose a starting point to make the tail sum $< \epsilon$. The choice of $L_N$ follows by taking
182     $\epsilon = N^{-(1+\delta)}$.

183     To obtain the choice of $D_N$, we first use a similar approach to the proof of Proposition 9 to simplify
184     $\tilde{\mathbb{T}}_{L,D'}$ for any $L, D'$:

$$\tilde{\mathbb{T}}_{L,D'} = \sum_{\ell=1}^{L} \left[ \frac{1}{N_1(N_1-1)} \sum_{i,j:i\neq j} \eta_{D'}(\phi_\ell \sharp \mu_i)^T \eta_{D'}(\phi_\ell \sharp \mu_j) + \frac{1}{N_2(N_2-1)} \sum_{i,j:i\neq j} \eta_{D'}(\phi_\ell \sharp \nu_i)^T \eta_{D'}(\phi_\ell \sharp \nu_j) \right.$$

$$\left. - \frac{2}{N_1 N_2} \sum_{i,j} \eta_{D'}(\phi_\ell \sharp \mu_i)^T \eta_{D'}(\phi_\ell \sharp \nu_j) \right]. \tag{A.8}$$

185 Recall that the inverse CDF transformation induced by $\eta_0(\phi_\ell \sharp \mu) \equiv F^{-1}_{\phi_\ell \sharp \mu}$ maps $[0,1]$ to a bounded
186 interval that is the range of $\phi_\ell$, and $\|\phi_\ell\|_\infty \le c\lambda_\ell^{(n-1)/4}$ using Hörmander's bound on the supremum
187 norm of the eigenfunctions. Using classical results on Riemann sum approximation errors [3, 6] we
188 thus have for any $\ell$:

$$\left| \alpha_\ell \langle \eta_0(\phi_\ell \sharp \mu), \eta_0(\phi_\ell \sharp \nu) \rangle_{\mathcal{H}} - \eta_{D'}(\phi_\ell \sharp \mu)^T \eta_{D'}(\phi_\ell \sharp \nu) \right| \le \frac{k}{D'} \alpha_\ell \left\| (F^{-1}_{\phi_\ell \sharp \mu} F^{-1}_{\phi_\ell \sharp \nu})' \right\|_\infty \le \frac{2kc^2}{D'} \alpha_\ell \lambda_\ell^{(n-1)/2}.$$

189 Given $L = L_N$, we simply choose $D' = D_N$ large enough to make the right hand side above smaller
190 than $N^{-(1+\delta)}$. While it is possible to make the upper bound tighter using recent results (such as [6]),
191 the above coarser bound suffices for our purpose. $\qquad\square$

192 We now state a version of Theorem 2 in the main paper, with specifications for $\gamma_m, \sigma_1^2, L_N, D_N$ now
193 available through the above two results.

194 **Theorem 2.** *Assume conditions (i)-(iii) hold. Define $N = N_1 + N_2$, and suppose that as $N_1, N_2 \to$*
195 *$\infty$, we have $N_1/N \to \rho_1, N_2/N \to \rho_2 = 1 - \rho_1$, for some fixed $0 < \rho_1 < 1$. With $L \ge L_N, D' \ge$*
196 *$D_N$ chosen per Proposition 10, under $H_0 : C_{\eta \# P} = C_{\eta \# Q}$ we have*

$$N\tilde{\mathbb{T}}_{L,D'} \rightsquigarrow \sum_{m=1}^\infty \gamma_m(A_m^2 - 1),$$

197 *where $A_m, \gamma_m$ are defined as in Proposition 9. Further, under $H_1 : C_{\eta \# P} \ne C_{\eta \# Q}$ we have*
198 *$\sqrt{N}\left(\tilde{\mathbb{T}}_{L,D'} - \mathbb{T}\right) \rightsquigarrow N(0, \sigma_1^2).$*

199 *Proof.* This a combination of Propositions 9 and 10, and Slutsky's theorem. $\qquad\square$

200 We conclude with a proof of Theorem 3, which gives power guarantee of the test based on $\tilde{\mathbb{T}}_{L,D'}$ for
201 contiguous alternatives.

202 **Theorem 3.** *Assume conditions (i)-(iii) hold, and let $L, D'$ be chosen as in Theorem 2. Then for*
203 *the sequence of contiguous alternatives $H_{1N}$ such that $N\|\delta_N\|_{\mathcal{H}}^2 \to \infty$, the test based on $\tilde{\mathbb{T}}_{L,D'}$ is*
204 *consistent for any $\alpha \in (0,1)$, that is as $N \to \infty$ the asymptotic power approaches 1.*

205 *Proof.* It is enough the prove consistency using $\hat{\mathbb{T}}$, as the difference between $\hat{\mathbb{T}}$ and $\tilde{\mathbb{T}}_{L,D'}$ is negligible
206 by choice of $L, D'$. To do so we utilize proof techniques similar to [12, Theorem 13]. Define
207 $c_N := N^{1/2}\|\delta_N\|_{\mathcal{H}}$, and expand the simplified centered version of the test statistic in (A.5) but under
208 $H_1$ so that the centering terms do not cancel out:

$$
\begin{aligned}
\hat{\mathbb{T}}_c = & \frac{1}{N_1(N_1 - 1)} \sum_{i,j:i \ne j} \langle \eta(\mu_i) - C_{\eta \# P}, \eta(\mu_j) - C_{\eta \# P} \rangle_{\mathcal{H}} \\
& + \frac{1}{N_2(N_2 - 1)} \sum_{i,j:i \ne j} \langle \eta(\nu_i) - C_{\eta \# Q}, \eta(\nu_j) - C_{\eta \# Q} \rangle_{\mathcal{H}} \\
& \left. - \frac{2}{N_1 N_2} \sum_{i,j} \langle \eta(\mu_i) - C_{\eta \# P}, \eta(\nu_j) - C_{\eta \# Q} \rangle_{\mathcal{H}} \right].
\end{aligned}
\tag{A.9}
$$

209 The centered pushforwards have the same Hilbert centroids, thus as $N \to \infty$ by Proposition 9,

$$N\hat{\mathbb{T}}_c \rightsquigarrow \sum_{m=1}^\infty \gamma_m(A_m^2 - 1) := S.$$

Subtracting $\hat{\mathbb{T}}_c$ from $\hat{\mathbb{T}}$ and its expansion in Eq. (A.2) on the left and right hand respectively, then simplifying we have

$$
\begin{aligned}
N(\hat{\mathbb{T}} - \hat{\mathbb{T}}_c) \quad = N \quad & \left[ -\frac{1}{N_1} \sum_{i=1}^{N_1} \langle \delta_N, \eta(\mu_i) - C_{\eta\#P} \rangle_{\mathcal{H}} + \frac{1}{N_2} \sum_{i=1}^{N_2} \langle \delta_N, \eta(\nu_i) - C_{\eta\#Q} \rangle_{\mathcal{H}} + \frac{\langle \delta_N, \delta_N \rangle_{\mathcal{H}}}{2} \right] \\
= N \quad & \left[ \frac{\|\delta_N\|_{\mathcal{H}}}{N_1} \sum_{i=1}^{N_1} \left\langle \frac{\delta_N}{\|\delta_N\|_{\mathcal{H}}}, \eta(\mu_i) - C_{\eta\#P} \right\rangle_{\mathcal{H}} \right. \\
& \left. - \frac{\|\delta_N\|_{\mathcal{H}}}{N_2} \sum_{i=1}^{N_2} \left\langle \frac{\delta_N}{\|\delta_N\|_{\mathcal{H}}}, \eta(\nu_i) - C_{\eta\#Q} \right\rangle_{\mathcal{H}} + \frac{\|\delta_N\|_{\mathcal{H}}^2}{2} \right].
\end{aligned}
\tag{A.10}
$$

Given $N$ the inner products $\langle \delta_N/\|\delta_N\|_{\mathcal{H}}, \eta(\mu_i) - C_{\eta\#P} \rangle_{\mathcal{H}}$ are i.i.d. random variables with mean 0, so by CLT then using $\|\delta_N\|_{\mathcal{H}} = c_N N^{-1/2}$ we get

$$
\frac{1}{\sqrt{N_1}} \sum_{i=1}^{N_1} \left\langle \frac{\delta_N}{\|\delta_N\|_{\mathcal{H}}}, \eta(\mu_i) - C_{\eta\#P} \right\rangle_{\mathcal{H}} \rightsquigarrow U \quad \Rightarrow \quad \frac{N\|\delta_N\|_{\mathcal{H}}}{N_1} \sum_{i=1}^{N_2} \left\langle \frac{\delta_N}{\|\delta_N\|_{\mathcal{H}}}, \eta(\nu_i) - C_{\eta\#Q} \right\rangle_{\mathcal{H}} \rightsquigarrow \frac{c_N}{\sqrt{\rho_1}} U,
$$

where $U$ is the zero mean Gaussian random variable that is the limiting distribution of the above inner product sum. Similarly we have

$$
\frac{N\|\delta_N\|_{\mathcal{H}}}{N_2} \sum_{i=1}^{N_2} \left\langle \frac{\delta_N}{\|\delta_N\|_{\mathcal{H}}}, \eta(\nu_i) - C_{\eta\#Q} \right\rangle_{\mathcal{H}} \rightsquigarrow \frac{c_N}{\sqrt{\rho_2}} V,
$$

where $V$ is also Gaussian, zero mean, and independent of $U$. Putting everything together in the right hand side of (A.10), and using $\|\delta_N\|_{\mathcal{H}} = c_N N^{-1/2}$, given the threshold $t_\alpha$ for a level-$\alpha$ test

$$
P_{H_N} \left( N\hat{\mathbb{T}} > t_\alpha \right) \rightarrow P \left[ S + c_N \left( \frac{U}{\sqrt{\rho_1}} - \frac{V}{\sqrt{\rho_2}} \right) + \frac{c_N^2}{2} > t_\alpha \right].
$$

By assumption $c_N^2 \rightarrow \infty$, so the asymptotic power approaches 1 as $N \rightarrow \infty$. $\qquad\square$

## A.5 Proofs and Notes for Section 4.2

To guarantee size control when using the the harmonic mean $p$-value we establish a version of Theorem 1 from [15]. Assume that a test statistic $Z \in \mathbb{R}^D$ has null distribution with zero mean and every pair of coordinates of $Z$ follows bivariate Gaussian distribution. Compute the coordinate-wise two-sided $p$-values $p_k = 2(1 - \Phi(|Z_k|))$ where $\Phi$ is the standard Gaussian CDF.

**Theorem 4.** *Let $p_k, k = 1, ..., D$ be the null $p$-values as above and $p^H$ computed via harmonic mean approach, then*

$$
\lim_{\alpha \to 0} \frac{\text{Prob}\{p^H \leq \alpha\}}{\alpha} = 1.
$$

*Proof.* The proof of Theorem 1 from [15] hinges on Lemma 3 in their supplemental material. We show that Lemma 3 holds for the harmonic mean combination method. Note that the multiplication by $\pi$ present in Lemma 3 cancels out when inverse cotangent with a multiplier of $1/\pi$ is applied later on; so it is not relevant to the flow of the proof.

To this end, consider the functions $p(x) = 2(1 - \Phi(|x|))$ and $h(x) = 1/p(x)$. We need to prove the following three statements:

(1) for any $|x| > \Phi^{-1}(3/4)$,

$$
\frac{\cos[p(x)\pi]}{p(x)} \leq h(x) \leq \frac{1}{p(x)}
$$

(2) For any constant $0 < |a| < 1$, we have

$$
\lim_{x \to +\infty} \frac{h(x)}{x^2 h(ax)} > c_a > 0,
$$

234 where $c_a$ is some constant only dependent on $a$.

235 (3) Suppose that $X_0$ has standard normal distribution, then we have

$$P\{h(X_0) \geq t\} = \frac{1}{t} + O(1/t^3).$$

236 Statement (1) is trivial, as $h(x) = 1/p(x)$ by definition and the cosine function is upper bounded by
237 one. Statement (2) holds by the same argument as in the supplement of [15]. Statement (3) follows
238 from the fact that when $X_0$ is standard normal, then $p(x)$ is a null $p$-value, and so

$$P\{h(X_0) \geq t\} = P\{p(X_0) \leq 1/t\} = \frac{1}{t}.$$

239 Note that there is no $O(1/t^3)$ term at all, but we kept the form of the statement the same as in [15].

240 Now, the proof of Theorem 1 from [15] with weights $\omega_k = 1/D, k = 1, 2, ..., D$ goes through to give

$$P\left\{\frac{1}{D} \sum \frac{1}{p_k} \geq t\right\} = \frac{1}{t} + o(1/t).$$

241 Note that $p^H = H\left(D/(\frac{1}{p_1} + \frac{1}{p_2} + \cdots + \frac{1}{p_D})\right)$, where the function $H$ has a known form described
242 in [26] and satisfies $H(x)/x \to 1$ as $x \to 0$. Thus, as $\alpha \to 0$, we have

$$P\{p^H \leq \alpha\} \asymp P\left\{\frac{1}{D} \sum \frac{1}{p_k} \geq 1/\alpha\right\} \asymp \frac{1}{1/\alpha} + o(\frac{1}{1/\alpha}) \asymp \alpha.$$

243 $\qquad\qquad\qquad\qquad\qquad\qquad\qquad\qquad\qquad\qquad\qquad\qquad\qquad\qquad\qquad\qquad\qquad\qquad\qquad$ □

## B  Details of numerical experiments

### B.1  Synthetic data

246 We compare the performance of our tests on data from a number of domains with several existing
247 methods, and settings of the embedding parameters $L, D'$. For evaluation, we use empirical power at
248 different degrees of departure from the null hypothesis, calculated by averaging the proportion of
249 rejections at level $\alpha = 0.05$ over 1000 independent datasets with samples divided into two groups
250 of sizes $n_1 = 60, n_2 = 40$. To ensure the tests are well-calibrated, we also calculate nominal
251 sizes assuming the two sample groups are drawn from the same random meta-distribution. We
252 calculate eigenvalues and eigenfunctions using analytical expressions provided in the Appendix, and
253 fix $\alpha(\lambda) = e^{-\lambda}$ (i.e. heat kernel with $t = 1$) for all experiments.

254 **Finite intervals**  To obtain our base measures $\mu_i, \nu_i$, we generate bin probabilities as (shifted
255 and normalized) values of the function $f(t_j) = \mu(t_j) + \alpha(t_j)$ at $m = 30$ fixed design points
256 $t_j = j/(m+1), j = \{1, 2, \ldots, m\}$, and

$$
\begin{aligned}
\mu(t_j) &= 1.2 + 2.3\cos(2\pi t_j) + 4.2\sin(2\pi t_j), \\
\alpha(t_j) &= \epsilon_0 + \sqrt{2}\epsilon_1 \cos(2\pi t_j) + \sqrt{3}\epsilon_2 \sin(2\pi t_j),
\end{aligned}
$$

257 where $\epsilon_0, \epsilon_1, \epsilon_2 \sim N(0, 1)$ clipped between $[-3, 3]$. Group 1 and 2 samples are obtained as $\mu_i(\cdot) \equiv$
258 $f(\cdot)$ and $\nu_i(\cdot) \equiv f(\cdot) + \delta$ respectively, where $\delta \in [0, 4]$ is a constant. To make the sample functions
259 non-negative, we shift all functions by $M = 3(1 + \sqrt{2} + \sqrt{3})$. Finally, as the $m$-length vector of
260 bin counts for a sample, we generate a random vector from the Multinomial distribution with 1000
261 trials, $m$ outcomes and the outcome probabilities proportional to the shifted functional observations
262 corresponding to that sample.

263 We use embedding dimensions $L = 3, D' = 10$ to compare our method against 11 functional ANOVA
264 tests—for brevity we report results for 3 of them which use different methodological approaches (see
265 Appendix for complete results). All methods maintain nominal size for $\delta = 0$ (Figure 1 a). While the
266 combination test (ISD comb) based on our proposal outperformed all the other tests across all values
267 of $\delta$, the bootstrap test that uses the overall $\mathbb{T}$ statistic (ISD T boot) performs better than Fmaxb but
268 worse than others. Table 2 shows the outputs for the other 8 competing methods from the R package
269 `fdANOVA` for the finite intervals synthetic data setting[1].

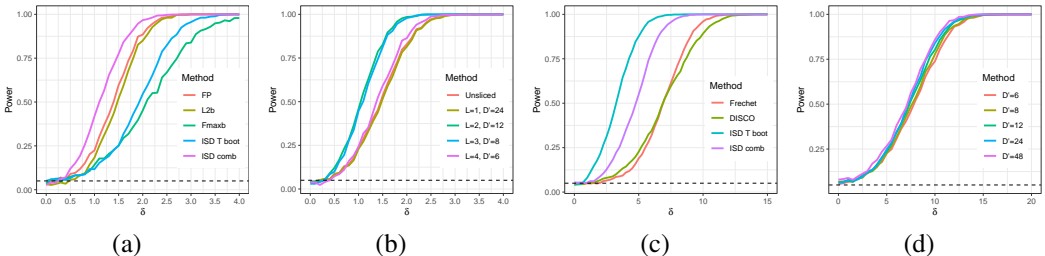

Figure 1: Performance on synthetic finite interval and manifold data. Finite interval: (a) comparison with existing methods—a test based on basis function representation (FP) [11], a sum-type $\ell_2$ norm-based test (L2b) [27], and a max-type test [28] that uses the maximum of coordinate-wise $F$ statistic (Fmaxb); (b) unsliced vs. different settings of $(L, D')$. Manifold data: (c) circular data, comparing with Fréchet ANOVA [9], and the DISCO nonparametric test [21]; (d) harmonic combination tests on cylindrical data for $L = 4$. Dotted lines indicates nominal size of all tests ($\alpha = 0.05$).

| $\delta$ | CH | CS | L2N | L2b | FN | FB | Fb | GPF |
|---|---|---|---|---|---|---|---|---|
| 0 | 0.031 | 0.03 | 0.033 | 0.024 | 0.031 | 0.028 | 0.033 | 0.026 |
| 0.1 | 0.025 | 0.024 | 0.03 | 0.044 | 0.027 | 0.03 | 0.041 | 0.021 |
| 0.2 | 0.026 | 0.029 | 0.037 | 0.06 | 0.033 | 0.034 | 0.058 | 0.025 |
| 0.3 | 0.036 | 0.041 | 0.044 | 0.067 | 0.041 | 0.04 | 0.067 | 0.033 |
| 0.4 | 0.034 | 0.035 | 0.036 | 0.057 | 0.034 | 0.035 | 0.056 | 0.032 |
| 0.5 | 0.051 | 0.052 | 0.058 | 0.091 | 0.056 | 0.057 | 0.088 | 0.044 |
| 0.6 | 0.056 | 0.066 | 0.066 | 0.089 | 0.061 | 0.066 | 0.088 | 0.051 |
| 0.7 | 0.07 | 0.083 | 0.083 | 0.121 | 0.084 | 0.081 | 0.119 | 0.064 |
| 0.8 | 0.085 | 0.097 | 0.095 | 0.151 | 0.093 | 0.094 | 0.144 | 0.081 |
| 0.9 | 0.118 | 0.142 | 0.14 | 0.2 | 0.144 | 0.137 | 0.194 | 0.118 |
| 1 | 0.158 | 0.182 | 0.176 | 0.232 | 0.183 | 0.173 | 0.228 | 0.154 |
| 1.1 | 0.215 | 0.247 | 0.246 | 0.303 | 0.251 | 0.242 | 0.301 | 0.212 |
| 1.2 | 0.27 | 0.31 | 0.303 | 0.375 | 0.311 | 0.3 | 0.368 | 0.27 |
| 1.3 | 0.328 | 0.363 | 0.357 | 0.438 | 0.37 | 0.353 | 0.43 | 0.324 |
| 1.4 | 0.395 | 0.432 | 0.432 | 0.504 | 0.436 | 0.423 | 0.499 | 0.394 |
| 1.5 | 0.488 | 0.52 | 0.514 | 0.592 | 0.521 | 0.511 | 0.586 | 0.483 |
| 1.6 | 0.534 | 0.595 | 0.576 | 0.652 | 0.593 | 0.566 | 0.647 | 0.544 |
| 1.7 | 0.628 | 0.677 | 0.669 | 0.723 | 0.678 | 0.661 | 0.719 | 0.631 |
| 1.8 | 0.704 | 0.737 | 0.727 | 0.789 | 0.748 | 0.725 | 0.785 | 0.707 |
| 1.9 | 0.785 | 0.823 | 0.812 | 0.869 | 0.827 | 0.806 | 0.867 | 0.793 |
| 2 | 0.83 | 0.849 | 0.844 | 0.88 | 0.85 | 0.841 | 0.875 | 0.832 |
| 2.1 | 0.865 | 0.888 | 0.881 | 0.916 | 0.887 | 0.878 | 0.915 | 0.872 |
| 2.2 | 0.903 | 0.922 | 0.916 | 0.946 | 0.928 | 0.912 | 0.946 | 0.907 |
| 2.3 | 0.938 | 0.95 | 0.944 | 0.964 | 0.951 | 0.944 | 0.963 | 0.944 |
| 2.4 | 0.958 | 0.973 | 0.967 | 0.977 | 0.972 | 0.966 | 0.976 | 0.964 |
| 2.5 | 0.974 | 0.98 | 0.976 | 0.985 | 0.981 | 0.975 | 0.985 | 0.974 |
| 2.6 | 0.977 | 0.981 | 0.979 | 0.987 | 0.981 | 0.978 | 0.986 | 0.977 |
| 2.7 | 0.989 | 0.996 | 0.992 | 0.997 | 0.996 | 0.992 | 0.997 | 0.991 |
| 2.8 | 0.997 | 0.998 | 0.997 | 0.998 | 0.998 | 0.997 | 0.998 | 0.996 |
| 2.9 | 0.996 | 0.997 | 0.996 | 0.999 | 0.997 | 0.996 | 0.999 | 0.997 |
| 3 | 0.998 | 1 | 0.999 | 1 | 1 | 0.999 | 1 | 0.999 |

Table 2: Outputs for other methods in the functional curves synthetic data setting.

We also compare the $p$-value combination test based on an *unsliced* 24-dimensional inverse CDF embedding with sliced $IS\mathcal{W}_2$-based tests (Figure 1 b). We use multiple pairs of $(L, D')$ values, all of them giving overall embeddings of dimension $D = LD' = 24$. The performance of an $IS\mathcal{W}_2$-based

---

[1]See https://www.rdocumentation.org/packages/fdANOVA/versions/0.1.2/topics/fanova. tests for full names of all methods.

test that uses slicing over only the first eigenfunction is almost as good as the unsliced version. With more eigenfunctions, the powers first improve considerably, then become similar to the unsliced version again.

**Manifold domains**   We consider data from distributions on circles and cylinders. For circular data, we take von Mises distributions with randomly chosen parameters as our samples. For an angle $x$ (measured in radians), the von Mises probability density function is given by $f(x|\mu, \kappa) = \exp[\kappa \cos(x - \mu)](2\pi I_0(\kappa))^{-1}$, where $I_0(\kappa)$ is the modified Bessel function of order 0. We fix $\kappa = 2$, and use $\mu \equiv \mu_i \sim N(0, 0.1^2)$, $\mu \equiv \nu_i \sim N(\delta, 0.1^2)$ for samples from group 1 and 2 respectively—with $\delta \in [0, 15] \times \pi/180$ (i.e. 0 to 15 degrees converted to radians). As each observation vector, we take 100 random draws from each sample-specific distribution. For our embeddings, we use $L = 10, D' = 20$, and so our final embedding dimension is $10 \times 20 \times 2 = 400$. Since the competing methods cannot handle circular geometry directly, to implement them we cut the circle into an interval. Figure 1 (c) shows that all methods maintain nominal size, but both our tests maintain considerably higher power than existing methods for all $\delta$.

We generate cylindrical data in the form of samples of a bivariate random vector $(\Theta, X)$, using the cylindrical density function proposed by [16]:

$$f(\theta, x) = \frac{e^{\kappa \cos(\theta - \mu)}}{2\pi I_0(\kappa)} \frac{1}{\sqrt{2\pi}\sigma_c} e^{-\frac{(x - \mu_c)^2}{2\sigma_c^2}},$$

clipping values of the $X$-coordinate between the bounded interval $[0, 2\pi]$. This distribution has the parameters $\mu \in [-\pi, \pi], \mu_0 \in \mathbb{R}, \kappa \geq 0, \rho_1 \in [0, 1), \rho_2 \in [0, 1), \sigma > 0$, where $\mu, \kappa$ denote parameters for the (circular) marginal along the $\Theta$-coordinate. and given $\Theta = \theta$, $X$ is sampled from $N(\mu_c, \sigma_c^2)$, with

$$
\begin{aligned}
\mu_c &= \mu + \sqrt{\kappa}\sigma \left\{ \rho_1(\cos\theta - \cos\mu) + \rho_2(\sin\theta - \sin\mu) \right\}, \\
\sigma_c &= \sigma^2(1 - \rho^2), \rho = (\rho_1^2 + \rho_2^2)^{1/2}.
\end{aligned}
$$

In our experiments, we fix $\rho_1 = \rho_2 = 0.5, \sigma = 1, \kappa = 2$ across both populations. As random samples of distributions, we draw $\mu, \mu_0 \sim \text{Unif}(0, 1)$ and $\mu, \mu_0 \sim \text{Unif}(\delta, \delta + 1)$ for samples of group 1 and 2 respectively, with $n_1 = 60, n_2 = 40$. We repeat the above for $\delta \in [0, 30]$ degrees converted to radians, and obtain bivariate histograms corresponding to each sample distribution from 500 random draws from that distribution. To evaluate the effects of choosing $L, D'$ we calculate our embeddings for $L \in \{2, 3, 4, 5\}, D' \in \{6, 8, 12, 24, 48\}$. The choice of $L$ has small effect on performance, so we report results for $L = 4$ in Figure 1 (d). Higher values of $D'$ result in some increase in power.

**Discussion**   Our $ISW_2$-based method is able to exploit the non-euclidean nature of the problems and and their generality beyond mean comparison more effectively than competing methods, which are based on mean comparison on functional data/densities (frechet ANOVA, all functional ANOVA methods), and/or L2 distance-based comparisons (all functional ANOVA methods, DISCO). Regarding the optimal choice of embedding dimensions, while proving theorem **??** we show that (Proposition 10 therein) choosing both $L$ and $D$ above certain thresholds ensures close approximation to the population test statistic. For the combination test, adding more dimensions to the embedding can have a two-fold effect: a) probing more dimensions can help with finding differences, but b) every dimension adds another test and so potentially leads to loss of power. Thus, for the combination test, there must be an optimal data dependent choice of the embedding dimension, which can potentially be found via split testing procedures. We leave this to future work.

## B.2   NHANES data on physical activity monitoring

As our first real data application, we analyze the Physical Activity Monitor (PAM) data from the 2005-2006 National Health and Nutrition Examination Survey (NHANES)[2]. This contains physical activity pattern readings for a large number of people collected over 1 week period on a per-minute granularity. After basic pre-processing steps to ensure no missing entries, as well as data reliability and well-calibrated activity monitors, we use data from 6839 individuals. The data for each individual

---

[2] https://wwwn.cdc.gov/Nchs/Nhanes/2005-2006/PAXRAW_D.htm

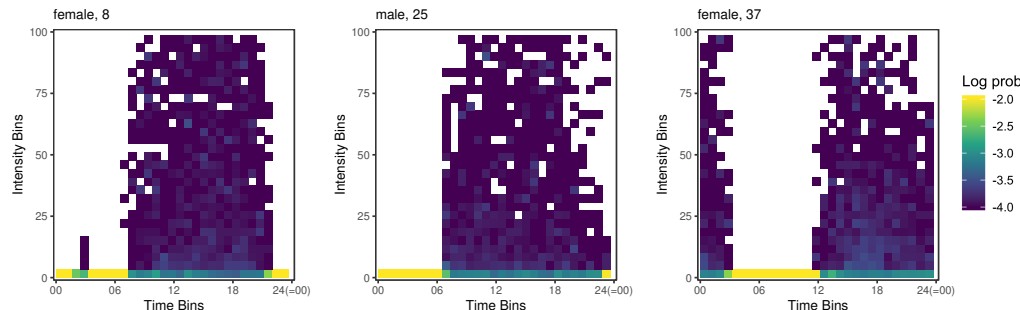

Figure 2: Activity histograms for three individuals from NHANES dataset. There are 100 bins in the intensity and 96 in the time dimension; we show hour of day on the time axis. The time dimension is periodic where 00:00 is identified with 24:00, giving rise to a cylindrical histogram domain.

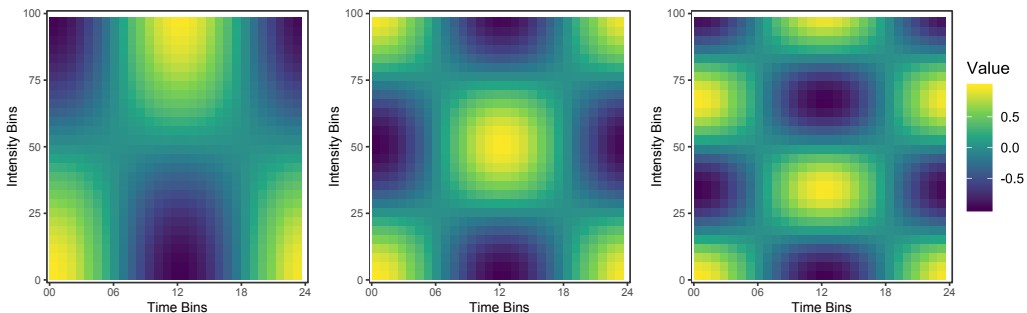

Figure 3: Three eigenfunctions for the NHANES histogram domain normalized by the maximum absolute value. Note that the eigenfunctions are periodic in the time direction (i.e. match when glued over the side cut) but not in the intensity direction, reflecting the cylindrical geometry of the underlying domain.

corresponds to device intensity value from the PAM for $24 \times 60 = 1440$ minutes throughout the day, for 7 days.

For each individual we can capture their activity patterns into a cylindrical histogram with time and intensity dimensions. For each observation, its time during the day is discretized into 15-minute intervals giving 96 bins for the time dimension; its intensity value (capped at 1,000) is discretized into a 100 equidistant bins. Since the time dimension is periodic, we obtain a histogram over the cylinder $S^1(T_1) \times [0, T_2)$, with $T_1 = 96, T_2 = 100$. Normalized counts can thus be considered as person-specific probability distributions; several examples are shown in Figure 2. Note that flattening the domain by cutting the cylinder will arbitrarily split activity patterns (see especially Figure 2, Female 37) and will lead to inefficiencies due to horizontal variability.

We apply the proposed methodology to check if the activity patterns vary across different groups of individuals obtained as follows. We first split the overall dataset based on the individual's age using the following inclusive ranges: 6–15, 16–25, ...,76–85; this covers all the ages in the dataset. From each split we sample 100 males and 100 females to avoid gender imbalance driving the results. Thus, we end up with 8 age groups with 200 individuals per group. Our goal is to compare these 8 groups' activity patterns by conducting pair-wise tests.

To perform our analysis we compute the eigenvalues and eigenfunctions as per the 4th row of Table 1 using $\ell_1 = 1, 2, 3$ and $\ell_2 = 1, 2, 3$, giving a total of $L = 2 \times 3 \times 3 = 18$ eigenfunctions; three of the resulting eigenfunctions are shown in Figure 3. We consider a $D' = 5$ dimensional embedding for the inverse CDF transformation, hence the final embedding dimension after the slicing construction is $D = LD' = 18 \times 5 = 90$.

We summarize the results in Table 3, *below the diagonal*. The $p$-values are obtained via the harmonic mean combination approach. We run the Benjamini-Hochberg [4] procedure on the resulting $p$-values

| Age Groups | 6–15 | 16–25 | 26–35 | 36–45 | 46–55 | 56–65 | 66–75 | 76–85 |
|---|---|---|---|---|---|---|---|---|
| 6–15 | | 0.979 | 0.31 | 0.383 | 0.297 | 0.905 | 0.921 | 0.326 |
| 16–25 | **3.7e-11** | | 0.998 | 0.963 | 0.443 | 0.872 | 0.442 | 0.529 |
| 26–35 | **4.6e-20** | **1.0e-05** | | 0.987 | 0.818 | 0.93 | 0.731 | 0.992 |
| 36–45 | **3.2e-26** | **3.5e-11** | **0.01** | | 0.945 | 0.984 | 0.974 | 0.327 |
| 46–55 | **6.6e-27** | **8.4e-16** | **0.002** | 0.377 | | 0.832 | 0.618 | 0.844 |
| 56–65 | **2.4e-32** | **7.5e-20** | **3.1e-04** | **0.042** | 0.977 | | 0.509 | 0.98 |
| 66–75 | **5.4e-45** | **1.6e-16** | **7.7e-06** | **1.6e-04** | **0.001** | **0.011** | | 0.557 |
| 76–85 | **3.4e-52** | **1.4e-23** | **1.4e-15** | **2.7e-12** | **9.7e-16** | **1.4e-09** | **2.1e-06** | |

Table 3: Comparing the activity intensity of different age groups based on the NHANES dataset. Below diagonal: $p$-values corresponding to the actual data comparisons. Above diagonal: null $p$-values obtained by combining and randomly splitting the two involved groups. The entries in boldface correspond to the rejected hypotheses with the BH procedure at the FDR level of 0.1.

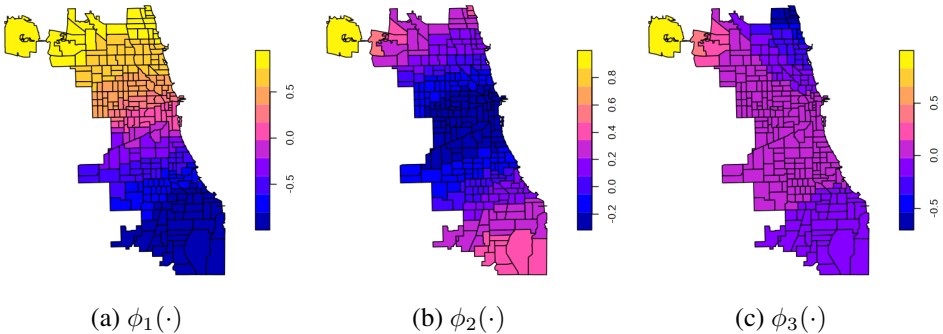

(a) $\phi_1(\cdot)$   (b) $\phi_2(\cdot)$   (c) $\phi_3(\cdot)$

Figure 4: First three eigenvectors of the Laplacian are shown for the beat adjacency graph, mapped back to the geographic locations. All of the eigenvectors are normalized by the maximum absolute value. The spatial smoothness of the eigenvectors—somewhat masked here due to the discrete colormap—is crucial to efficiently capturing horizontal variability of the data (i.e. distribution shifts over the graph). The boundaries of beats are shown based on the shape file from Chicago Data portal.

at the false discovery rate of $0.1$, and the rejected hypotheses are indicated by the $p$-values in bold. Our method detects statistically significant differences between all pairs of groups, except 46–55 versus 36–45 and 56-65 groups. As a control experiment, we provide our method with null cases and display the $p$-values in Table 3, *above the diagonal*. The null cases are obtained by combining the individuals from the two comparison groups and splitting it arbitrarily (i.e. mixing the two age groups). As expected, the $p$-values of the control comparisons do not concentrate near zero.

Curiously, our method can be used "off-label" to conduct *functional data analyses* over different dimensions of the NHANES dataset. For example, one can concentrate on a single day of activity intensity data which gives a curve over the 24-hour circle. Since activity intensity is a non-negative number, these curves can be normalized so as to obtain probability distributions. Now we can use our methodology to detect pair-wise differences across groups. While this has the benefit of accounting for underlying geometry of data, it loses the absolute magnitude information due to the normalization. Clearly the appropriateness of such an analysis would depend on the goal of the exercise and the particular research question attached to that goal; our proposal provides a framework that is flexible enough to handle data of different modalities.

### B.3 Chicago Crime

We demonstrate the use of our methodology on histograms over graphs. In this experiment, we use the Chicago Crimes 2018 dataset[3] which captures incidents of crime in the City of Chicago. We base our analysis on the type of crime, the beat (geographic area subdivision used by police, see Figure 4)

---

[3] `data.cityofchicago.org`

| Crime Type | Tuesday | | Thursday | | Saturday | | Tue vs Thu | Tue vs Sat |
|---|---|---|---|---|---|---|---|---|
| | $N$ | $\overline{\text{count}}$ | $N$ | $\overline{\text{count}}$ | $N$ | $\overline{\text{count}}$ | $p$-value | $p$-value |
| Theft | 52 | 178.7 | 52 | 182.9 | 52 | 180.2 | 0.452 | **4.7e-06** |
| Deceptive Practice | 51 | 55.8 | 52 | 54.9 | 52 | 44.4 | 0.255 | **4.2e-04** |
| Battery | 52 | 125.8 | 52 | 123.0 | 52 | 154.9 | 0.374 | **0.001** |
| Robbery | 50 | 25.2 | 50 | 25.1 | 52 | 28.1 | 0.130 | **0.002** |
| Narcotics | 51 | 36.0 | 51 | 34.6 | 50 | 36.9 | 0.890 | **0.008** |
| Criminal Damage | 52 | 70.0 | 52 | 73.7 | 52 | 83.0 | 0.901 | **0.03** |
| Other Offense | 52 | 49.5 | 52 | 48.4 | 52 | 44.1 | 0.670 | 0.037 |
| Burglary | 52 | 34.0 | 52 | 33.1 | 52 | 29.1 | 0.157 | 0.183 |
| Motor Vehicle Theft | 52 | 27.9 | 52 | 26.2 | 51 | 28.1 | 0.923 | 0.365 |
| Assault | 52 | 57.2 | 52 | 59.3 | 52 | 52.4 | 0.996 | 0.617 |

Table 4: Results on Chicago Crime 2018 dataset. The entries in bold correspond to the rejected hypotheses with the BH procedure at the FDR level of 0.1. The $N$ column captures the number of days passing the filtering criteria, and the $\overline{\text{count}}$ column shows the average per-day crime count.

where the incident took place, and the date of the incident. To capture the spatial aspect of the data we build a graph with one vertex per beat; two vertices are connected by an edge if the corresponding beats share a geographic boundary. For each crime type and day, we capture the total count of that crime type for each beat; after normalizing this gives a daily probability distribution over the graph. Our goal is to compare the collection of distributions of, say, theft occurring on Tuesday to those of Thursday and Saturday. The Tuesday versus Thursday comparison is intended as a null case, as we do not expect to see any differences between them [23].

We build the un-normalized Laplacian of the beat adjacency graph, and compute its lowest frequency $L = 20$ eigenvalues and eigenvectors. The first three eigenvectors are plotted in Figure 4. The number of inverse CDF values used in the embedding is $D' = 5$, which gives rise to $D = 100$ dimensional embedding. The results of comparisons are shown in the last two columns of Table 4; the $p$-values are obtained via the harmonic mean combination approach. We run the Benjamini-Hochberg [4] procedure on the 20 resulting $p$-values at the false discovery rate of 0.1, and the rejected hypotheses are indicated by the $p$-values in bold. As expected, no differences were detected between Tuesday and Thursday patterns. On the other hand, we see that there are statistically significant differences between Tuesday and Saturday patterns in the following categories of crime: theft, deceptive practice, battery, robbery, narcotics, and criminal damage.

## B.4 Brain Connectomics

In this example, we consider two publicly available brain connectomics datasets [1, 2] distributed as a part of the R package `graphclass`[4]. Both are based on resting state functional magnetic resonance imaging (fMRI): COBRE has data on 54 schizophrenics and 70 controls, and UMich with 39 schizophrenics and 40 controls. The datasets capture the pairwise correlations between 264 regions of interest (ROI) of Power parcellation [18] and can be considered as a 264 node graph (263 nodes for COBRE as ROI 75 is missing) with positive and negative edge weights.

We define three probability measures supported on the nodes of the graph. For each ROI we take the sum of absolute values of all its correlations with the remaining ROIs. Now we have a positive number assigned to each node capturing its overall connectivity to the rest of the graph and we normalize to obtain a measure; this construction will be referred to as "all correlations". Note that each scanned subject gives rise to a separate "all correlations" probability measure on the same underlying node set. The "positive correlations" and "negative correlations" constructions are based on keeping respectively only positive or only negative correlations and aggregating as above.

We also need a fixed base graph for the computation of the Laplacian eigen-decomposition; this graph should capture the spatial connectivity of the ROIs which is relevant due to the smooth nature of the blood oxygenation level dependent (BOLD) signal that is used for computing the correlations.

---

[4]http://github.com/jesusdaniel/graphclass

| Dataset | All correlations | Positive correlations | Negative correlations |
|---------|------------------|-----------------------|-----------------------|
| COBRE | 0.0084 | 0.00019 | 0.0019 |
| UMich | 0.609 | 0.116 | 0.022 |

Table 5: Comparison results between the schizophrenic and control groups for brain connectomics datasets.

To this end, we obtain the coordinates for the centers of the 264 ROIs[5] and build the base graph by connecting each ROI to its nearest 8 ROIs.We compute the lowest frequency $L = 20$ eigenvalues and eigenvectors of the corresponding un-normalized Laplacian. The number of inverse CDF values used in the embedding is $D' = 5$, which gives rise to $D = 100$ dimensional embedding.

Table 5 shows the result of comparing the schizophrenic group to the control group for both of the datasets; the $p$-values are obtained via the harmonic mean combination approach. We can see that our approach detects statistically significant differences between the two groups in COBRE dataset in all of the three types of measures on graphs. In contrast, for UMich dataset, the difference is detected only in the negative correlations and loses significance when corrected for multiple testing. This is potentially caused by the higher inhomogeneity of the UMich dataset that was pooled across five different experiments spanning seven years [2]. An interesting aspect of our analysis is that due to normalization (to obtain probability measures) the total sum of connectivity is factored out by the proposed method. As a result, the detected differences are not related to the well-known change in the overall connectivities between the two groups, but rather to distributional changes in marginal connectivity strengths.