# OpenReview forum: "Intrinsic Sliced Wasserstein Distances for Comparing Collections of Probability Distributions on Manifolds and Graphs"
_NeurIPS.cc/2022/Conference — NeurIPS 2022 Submitted_

### Official Review · Reviewer_tW4F · 2022-06-29

**Rating:** 4
**Confidence:** 4
**Soundness:** 2 fair
**Presentation:** 3 good
**Contribution:** 3 good

**Summary:**

Consider the set of probability measures on some manifold. The authors propose a new distance on this set, that behaves in some way like the sliced Wasserstein distance. This distance is defined as the weighted average of the Wasserstein distance between the pushforwards of the two measures by the eigenfunctions of the Laplace-Beltrami operator on the manifold. Some properties of this distance are given, and the distance is then used in the two-sample testing problem, where each sample is a sample of iid probability measures on a given manifold. Some numerical illustrations of the two-sample tests are then given.

**Questions:**

On the implementation:
In a lot of situations, the measures at stake will be atomic, of the form $\mu_i= \sum a_i \delta_{x_i}$. Then, it looks like the most efficient way to compute the ISW distance is to compute $\phi_k(x_i)$ for every eigenfunction $\phi_k$, and then compute the 1D matching between two 1D atomic distributions (that is we do the same as for the classical sliced Wasserstein distance). Why is this approach not at all proposed in the paper?

On the numerical illustrations:
The authors test their procedures against other reasonable testing procedures, but not against the one that appears to be the most standard in their situation, that would be a two-sample test based on the Wasserstein distance. (that is we reject if $W_2(\mu_n,\nu_n)>\alpha$, and we accept otherwise). CLTs for empirical Wasserstein distances could probably be used to obtain asymptotic confidence levels on such a test, or one could use bootstrap or other approaches. Is there a reason for which this was not implemented (at least for synthetic experiments where the number of observations is reasonably small, so that W2 should be easy to compute)?

**Limitations:**

Yes.

**Strengths And Weaknesses:**

Sliced Wasserstein (SW) distances have been originally proposed as computationally cheap versions of the Wasserstein distance on the Euclidean space $\mathbb{R}^D$. The idea of the original SW is the following: instead of computing the Wasserstein distance between two measures in high dimension, let us look at a certain number of "projections" of the measures on lines in $\mathbb{R}^D$; then one can compute efficiently a distance between the two measures by averaging the distance between the projections.  This paper proposes a very reasonable generalization of this idea, called the Intrinsic SW (ISW) when the support of the measure is a manifold or a graph: as lines are no longer defined, the distance is defined as an average of "projections" along the eigenvalues of the Laplace-Beltrami operator, that should describe the geometry of the manifold (and therefore replace the lines in $\mathbb{R}^D$).

The paper focuses on the notion of ''Hilbert embeddability'' to justify the use of this distance. This means that their distance $ISW(\mu,\nu)$ is actually equal to $\|\eta(\mu)-\eta(\nu)\|_H $ for some feature map $\eta$ and some Hilbert space $H$. However, the authors seemed to have miss a key link with kernel methods. Indeed, the feature map $\eta$ defines a kernel on the space $P(X)$ of probability measures on $X$ (through $k(\mu,\nu) = <\eta(\mu),\eta(\nu)>_H$), and the method they propose for the two-sample tests is the kernel two-sample test method from Gretton & al. The authors seemed to have understand partially the connection, as their proofs are adaptations of the ones from Gretton & al, but the authors have missed that they could actually directly used the results from Gretton & al, without any adaptation.

There are other concepts defined in the paper that can be rephrased in the more standard language of kernels:
- a Hilbert centroid (def. 1) is a kernel mean embedding.
- T(P,Q) (eq 2.1) is actually the MMD associated with the kernel on P(X) given by the intrinsic sliced Wasserstein distance.
- Most proofs corresponding to Section 4.1 can be removed: it suffices to check that the assumptions given in Gretton & al are satisfied for the kernel given by the intrinsic SW distance.

Other comments:
- The intrinsic SW distance depends on the choice of a weight function (describing the importance given to each eigenvector). Having to tune such an important hyperparameter is clearly a drawback of this distance compared to the Sliced Wasserstein distance.
- Notation in Prop.9 (in the Appendix) are extra confusing: when defining the eigenvectors of some operator (that are defined on some Hilbert space H), the variable "x" is used, whereas x was used before to note an element of the underlying manifold. As we also consider eigenvectors on the manifold, this created a lot of confusion at first.

On my grade: I put 4 so far, as I believe the link with kernel methods is not made very clear in the paper, whereas it is central to the approach proposed. I could improve my grade if this link is made more explicit (e.g. say explicitly that the test IS a kernel two-sample test and that $T(P,Q)$ IS a MMD), and if proofs are significantly shortened by using Gretton & al.'s theorems directly. Please tell me if I am missing something and if this test is not a kernel two-sample test.

---

> ### Author Response · Authors · 2022-08-02
> **Differences from MMD**
>
> Thanks for the deep dives in your comments. Please see our clarifications below.
>
> **Is the Hilbert centroid (def. 1) a kernel mean embedding?**
>
> Not quite. While for each measure $\mu\in\mathcal{P}(\mathcal{X})$, we can think of $\eta(\mu)$ as a kernel mean embedding, our focus
> in this paper is on meta-distributions $P\in\mathcal{P}(\mathcal{P}(\mathcal{X}))$. The Hilbert centroid is defined for meta-distributions, and is basically mean (more precisely, centroid) of $\eta(\mu)$ where $\mu\sim P$. In other words, Hilbert centroid is the mean of kernel-mean-embeddings of distributions drawn from the meta-distribution $P$.
>
> **Is $T(P,Q)$ (eq 2.1) actually the MMD associated with the kernel on $P(X)$ given by $ISW_2$?**
>
> Not quite. Eq 2.1 is basically the formula for the squared distance between centroids of two distributions. For this to become an MMD
> one needs another kernel, say Gaussian kernel $e^{-x^{2}/2}$ applied on top of $D(\cdot,\cdot)$, and also flip the overall sign of the formula. Now testing using this quantity would be equivalent to MMD; and the induced kernel mean embedding will apply to meta-distributions.
>
> So, why not use MMD instead? First, there is a theoretical issue---the null hypothesis of this test is that $\eta\\#P=\eta\\#Q$, but there is no mathematical theory to imply that this is equivalent to $P=Q$. This raises a host of issues for the MMD theory which we do not know how to answer. Second, when tests like MMD reject the null, they do not provide any clues about what aspect(s) of two distributions are different; our proposed approach of concentrating on equality of centroids is more interpretable because one can drill down into the eigenfunction dimensions that display differences.
>
> **Is it possible to simplify/remove most proofs corresponding to Section 4.1 by checking if the assumptions given in Gretton et al are satisfied for the kernel given by $ISW_2$?**
>
> Not quite, as follows from the points above. Additionally, our null hypothesis is $H_{0}:C_{\eta\\#P}=C_{\eta\\#Q}$ which is different
> from MMD null hypothesis of $H_{0}:P=Q$. The latter is a much stronger assumption that simplifies the computation of the null distribution
> in the MMD case. In our case, we have to resort to a mixture distribution $R$ (see Proposition 9 in Appendix) instead of just using either
> $P$ or $Q$ (as they are equal under the MMD null), which requires a different technique for the second half of the proof. Even in terms
> of practical computation, note that MMD testing relies on permutation null which is valid due to the stronger null hypothesis of $H_{0}:P=Q$. In contrast, with our null hypothesis we cannot use the permutation null and have to resort to a bootstrap procedure.
>
> **The choice of a weight function being a drawback for $ISW_2$**
>
> We think of this freedom as of an advantage of our approach -- providing diversity of distance measures that can be tailored to the application needs. Different weight profiles are connected to some known distances such as diffusion and biharmonic distance and can be directly used by a practicioner.
>
> We note that when using the p-value combination test, these weight functions cancel out; this happens because the individual t-tests
> divide by the variance of the coordinate which cancels out the constant scaling of each coordinate by the weight function.
>
> **Confusing notation in Prop.9 in the Appendix**
>
> Thanks for pointing this out! We will clarify in these proofs that x refers to elements of the Hilbert space.
>
> **Computation for atomic measures**
>
> If one is only interested in computing only the $ISW_2$ distance, one can use 1D matching method as you mention. Our computation is based on the formula for the 2-Wasserstein distance in terms of the inverse CDF (Eq. 2.2), which leads to the discretized embedding formula of Eq 3.1 and eventually the formula for $\eta_{D}$. This embedding formula is crucial to the focus of this paper which is on testing. The embeddings can also be seen as alternate representations of manifold-valued data, other applications of which we plan to explore in future. For precise computation of ISW one can also use Eq 2.2/Eq 3.1 by discretizing at the atom locations which will give exactly the same result as 1D matching; however, due to different discretization locations, $\eta_{D}$ would not be consistently defined and would not be suitable for testing and other downstream uses.
>
> **Comparison against two-sample test based on the Wasserstein distance**
>
> Wasserstein distances can be used to test the equality of two usual distributions from $\mathcal{P}(\mathcal{X})$, but we are unaware
> of any such test for comparing meta-distributions from $\mathcal{P}(\mathcal{P}(\mathcal{X}))$. Having said that, we do present comparisons of different versions of intrinsic slicing with (unsliced) $W_2$ in the real interval example (Fig. 2b). We shall clarify this in the paper.

---

> > ### Comment · Reviewer_tW4F · 2022-08-09
> > **Further explanation?**
> >
> > I might be missing something here, but I still do not get why this is not a $T(P,Q)$ is not a MMD.
> > - Define a kernel $k$ on $X'=\mathcal{P}(X)$ by $k(\mu,\nu)=\langle \eta(\mu),\eta(\nu)\rangle$. Note that this is a kernel on $X'$ and not on $X$.
> > - The kernel mean embedding associated with a kernel $k$ (and associated feature map $\eta$) is by definition a function that takes as an input a probability measure $P$ on $X'$ (so here this is what is called a meta-distribution), and outputs $\int \eta(x')d P(x')$, which is exactly $C_{\eta \sharp P}$, and what is called a Hilbert centroid here. Therefore, a Hilbert centroid is a kernel mean embedding, but for a kernel defined on $X'=\mathcal{P}(X)$.
> > - Likewise, $T(P,Q)$ is the distance between the kernel mean embeddings, and therefore is a MMD (once again, the kernel is defined on $X'=\mathcal{P}(X)$, so that the MMD is a distance on $\mathcal{P}(X')=\mathcal{P}(\mathcal{P}(X))$).
> >
> > Could you further elaborate on why the Hilbert centroid is not a kernel mean embedding and why $T(P,Q)$ is not a MMD? (once again, I agree that this not a MMD on $\mathcal{P}(X)$ but on $\mathcal{P}(\mathcal{P}(X))$.)

---

> > > ### Author Response · Authors · 2022-08-09
> > > **MMD explanation**
> > >
> > > Thank you for the clarification. As you suggest, one can define this as Kernel mean embedding (KME). However it is not useful because the main property of KME used in the MMD theory is that $KME(P) = KME(Q) \Rightarrow P = Q$. This is central to the MMD theory, and the respective proofs rely on this property (e.g. results in [Gretton et al, 2012](https://www.jmlr.org/papers/volume13/gretton12a/gretton12a.pdf)). In our case this does not hold. Not only that, it is not even true that $KME(P) = KME(Q) \Rightarrow \eta \\# P=\eta\\# Q$ holds, because the underlying kernel is basically a linear kernel. Since $KME(P) \equiv C_{\eta \\# P}$, KME equality implies equality of the hilbert centroids, not the underlying distributions. Thus, while technically correct, the MMD approach does not lead to any benefits, eg we cannot use MMD theory to derive our theorems, rather it complicates things. Essentially, for the MMD theory and results from Gretton et al to be applicable, we would need to use an additional kernel on top of the current embedding (see our earlier response) and then derive and verify additional conditions under which $\eta \\#P = \eta \\# Q \Rightarrow P = Q$. We grappled with this issue in the early stages of our work, but later were able to gain some intuition that the above implication is not always true; however, we do not have a counterexample at this point.
> > >
> > > To take it to extreme, one can claim that every mean is KME and every distance is MMD, e.g. consider testing for mean equality of two distributions in $\mathbb R^n$. We can say that $x \mapsto x$ is a feature map, and so the mean vector is a KME, and the distance between means is MMD. There is no benefit in this, as the MMD theory does not apply, e.g. KME equality does not imply that the distributions are the same, it only means their means are the same.

---

### Official Review · Reviewer_ib9Y · 2022-07-08

**Rating:** 6
**Confidence:** 4
**Soundness:** 3 good
**Presentation:** 3 good
**Contribution:** 2 fair

**Summary:**

It is well-known that the Wasserstein distance between two 1D measures can be computed easily. Hence, a common approach in a higher dimensional Euclidean space $\mathbb{R}^d$ consists in _slicing_ the space (formally, projecting on 1D subspaces), compute the distances on the slices, and integrating to retrieve a proper distance between the input measures.

This work proposes to adapt this idea in the context of measures supported on a (compact) manifold $\mathcal{M}$. The key idea is to rely on the Laplacian-Beltrami operator on the manifold, which provides a collection of maps $\phi_\ell : \mathcal{M} \to \mathbb{R},\ \ell \in \mathbb{N}$ (eigenvectors of the operator), which can be used to push-forward the measures we want to compare to $\mathbb{R}$. Namely, they introduce
$$ISW_2(\mu,\nu) = \sum_\ell \alpha(\lambda_\ell) W_2(\phi_\ell \mu, \phi_\ell \nu), $$
where $\lambda_\ell$ is the eigenvalue of the Laplacian operator, $\alpha$ is some weighting function.

Authors then prove that $ISW_2$ defines a metric (under mild assumptions), the key aspect being that it induces a universal kernel which yields the separation property $ISW_2(\mu,\nu) = 0 \Rightarrow \mu = \nu$ (other properties are mostly straightforward).

Once this distance is introduced, authors provide some approximation results for practical computations (e.g. we do not have access to all the $(\phi_\ell)_\ell$ and similarly just a sampling of the (projected) measures. Afterward, as this metric is _Hilbertian_, meaning that the metric space $(P(X), ISW_2)$ isometrically embeds in a Hilbert space, they design a two-sample-test between distribution supported on $P(X)$ whenever $X$ is a compact manifold.
They showcase the relevance of their approach in different numerical experiments.

(note : $\phi \mu$ denotes the pushforward for $\mu$ by $\phi$ here, hashtag not used due to rendering issues.)

**Questions:**

# Questions

1. In the Chicago data experiment, why is the manifold/graph structure "crucial" as stated in the Caption of Figure 4 in the appendix ? i guess that we can reasonably assume that this graph is planar in which case we can consider the measures are supported on $\mathbb{R}^2$. As far as I can tell, there is no comparison between $ISW_2$ and $W_2^{\mathbb{R}^2}$ (or a standard sliced-$W_2$, so that one has a kernel embedding and can run a two-sample-test here).
2. Related to the comparison between the $ISW_2$ and the usual Sliced-Wasserstein (assuming our manifold is embedded in $\mathbb{R}^n$, what is the computational burden in terms of running time related to the computation of $ISW_2$ vs $SW_2$ (in situations where we do not have access to the $\phi_\ell$ and $\lambda_\ell$ in close form)? I think a proper comparison of the benefits of using $ISW_2$ instead of a naive $SW_2$ is worth of interest.

# Minor remarks and suggestions

- I think there are few typos in the Proof of Proposition 3 (some $\mathcal{H}$ should be $\mathcal{H}^0$, or $\eta$ should be $\eta^0$).
- The proof of Theorem 1 in the appendix could be enhanced (with some more precise quote of the results used in the aforementioned references). E.g. what precise statement in [17] and [12] yield the implication $\alpha(\lambda_\ell) > 0 \Rightarrow$ the kernel is universal hence characteristic? Right now, it is cumbersome to properly check the proof. A similar comment holds for other proofs as well (e.g. the proof of Theorem 2 is quite expeditious).
- In think the quantitative results mentioned in section 4.1 and detailed in the appendix should actually be mentioned in section 3.2 directly. Indeed, when reading section 3.2 at first, one may be worried by the lack of quantitative assertion there.
- line 325 : inite-->finite (typo).

**Limitations:**

Few limitations of the work should be discussed further.

I did not identify ethical concern specific to this work.

**Strengths And Weaknesses:**

# Originality

Using the Laplace-Beltrami operator as a way to build a "canonical" way to slice measures supported on a manifold feels pretty satisfying and is a novel idea to the best of my knowledge.
On the other hand, one could argue that once the idea of using the Laplace-Beltrami operator to slice the measures has been introduced, most of the work (aside from the experimental section) is essentially an adaptation of standard techniques in two-sample-tests literature.

# Clarity

The paper was overall well-written. It motivates the need to consider that our measures are supported on manifold in a convincing manner (e.g. on a circle for daily event records) and aside from few paragraphs (see below), it is fairly easy to read.

# Significance

I think this paper provides a new interesting approach that may motivate further practical developments of Optimal Transport for measures supported on non-flat domain.

# Quality

I think this is a competent paper which introduces a new idea that may be of interest for the Optimal Transport community.

---

> ### Author Response · Authors · 2022-08-02
> **Originality, graph structure, complexity**
>
> We appreciate your enthusiasm about the paper. Please see our responses below.
>
> **Isn't this method essentially an adaptation of standard techniques in two-sample testing literature?**
>
> Due to the attractive and general definition of ISW, this may seem to be the case at first glance. However we would like to stress that
> our meta-distribution testing framework on general domains is an important contribution, as evidenced by the fact that previous works that capture limited settings of this problem (such as interval/circle) published at top statistical/ML venues (e.g. ref [5] and other papers on functional and object data analysis). Our approach has unique features that make it very different from standard two-sample testing; for more in-depth discussion of these differences---especially with MMD---please see our response to Reviewer tW4F.
>
> **Is the manifold/graph structure "crucial"?**
>
> While this does not directly apply to the Chicago crime example, one can easily construct examples where two nodes of a planar graph are located close to each other in Euclidean distance, but the road network does not link them directly (e.g. if there is a river separating towns A and B, and so while they are close, the actual drive takes a long time as it has to pass through a brigde that's far away). In this
> case, approaches based on Euclidean projection will not be able to succesfully capture the topology of the underlying domain to widely
> separate A and B, whereas intrinsic approach will be able to do so. This is important for testing, as any shifts of probability mass from
> A to B will be seen as having small transporation cost in Euclidean distance.
>
> **Computational burden ISW vs SW**
>
> The computation of eigenvectors/eigenvalues of symmetric sparse matrices is a well studied problem with stable and efficient algorithms available, e.g. ARPACK providing an implementation of the Arnoldi method that can be interfaced from MATLAB, Python, or R. These methods incur linear time complexity in the size of graph (more precisely, $O(k(|V|+|E|))$ for $k$ eigenvectors (=intrinsic slices) of the graph Laplacian). This computational overhead is negligible (<<1 second) for the graph experiments in this paper. As a contrast, Sobolev transport which is one of the methods Reviewer LPx3 mentions for comparison, requires computing the Sliced Sobolev Transport Distance. This depends on the computation of all shortest paths from $k$ randomly selected nodes in a graph, which has complexity $O(k(|V|\log|V|+|E|))$ (see page 4 of their [paper](https://arxiv.org/abs/2202.10723)).
>
> Please refer to the response Reviewer LPx3 above for empirical comparison of computation times of our method with existing slicing techniques in the literature.
>
> **Eigenvector computation**
>
> This computation is done only once for the underlying domain and can be used to test any number of distributions on this domain. The rest of computation has the same complexity order for SW and ISW. Note that SW can have its own inefficiencies, e.g. when the manifold has low intrinsic dimensionality but is embedded in a high dimensional space, most of the SW projections are correlated or uninformative, which incurs not only compute cost but also leads to lower testing power.
>
> **Typos, roof of Proposition 3, Appendix could be enhanced**
>
> Thank you for pointing these out! We will fix the typos and the proof. We will add more details to the proofs in the Appendix.
>
> **Mention results of Section 4.1 in Section 3.2**
>
> We have re-organized this paper many times to help with readability! While we are open to further suggestions, our thinking is that Section
> 3.2 is broadly accessible and provides a straightforward description of the resulting approach via formula for $\eta_{D}$ that can be
> directly used for implementation.

---

> > ### Comment · Reviewer_ib9Y · 2022-08-09
> > **Thanks**
> >
> > Thank you for taking time answering my comments.
> >
> > > While this does not directly apply to the Chicago crime example, one can easily construct examples where two nodes of a planar graph are located close to each other in Euclidean distance
> >
> > I completely agree with this idea, but then it indeed makes the Chicago Crime dataset not very suited to showcase the strength of your approach. Basically, I was just wondering "what would make the Euclidean distance between cells bad here ?".
> > If you can work out another such example, that would be an instructive addition to the work.

---

> > > ### Author Response · Authors · 2022-08-09
> > > **Graph example**
> > >
> > > We constructed the example of a path graph embedded on the plane in the response to Reviewer LPx3 above from exactly this motivation. In this example, our ISW embeddings demonstrate superior performance compared to existing techniques---please see our comments above.

---

### Official Review · Reviewer_LPx3 · 2022-07-11

**Rating:** 4
**Confidence:** 4
**Soundness:** 2 fair
**Presentation:** 2 fair
**Contribution:** 2 fair

**Summary:**

The authors propose the intrinsic sliced Wasserstein (ISW) distances for collections of probability measures. The ISW is a variant of sliced Wasserstein (SW) which exploits the closed-form solution of optimal transport on 1-dimensional space. The authors propose to use the eigenfunctions to project supports into 1-dimensional space and use corresponding eigenvalues as its weights.

ISW is Hilbert embeddable which allows to reformulate testing problems over collections of probability measures to traditional ones in Hilbert space. The authors demonstrate it for hypothesis testing via resampling based test and testing by p-value combination (from coordinate-wise tests).


**Questions:**

+ The proposed ISW is a variant of sliced-Wasserstein (SW) by leveraging the eigenfunctions and eigenvalues (closely related to SW, GSW, TSW as in line 68-80; and the recent Sobolev transport – a variant of optimal transport for measures supported on graphs). The authors should compare them in experiments.
+ The motivation of using eigenfunctions (for slicing) is too vague (the authors only give a short discussion in line 179-182). I think the authors should describe its advantage rigorously in theory and/or experiments.

Some other concerns are as follow:
+ What is the relation between T in Equ. (2.1) and in Proposition 1?
+ For Proposition 6, is it hold for any kernel used in MMD? Please specify it.
+ In Proposition 8, why can one assume that C is finite? (it is not clear whether the bound is useful in Proposition 8 without this assumption)
+ For line 266-269, how to choose L, D’ to obtain infinitesimally small error estimation (e.g., to obtain error less than epsilon)?
+ For experiments, the authors should discuss the results, e.g., how to choose L, D’ (in Figure 2b) follow the results in Section 3.2 and Section 4?

Minor:
+ In line 193-194, \eta^0 maps to H^0, but D is a norm on H’?


**Limitations:**

+ The authors have not discussed the limitations of their works clearly in the text. (as mentioned in the checklist).
+ The authors have not discussed the potential negative societal impacts of their work as in the checklist.


**Strengths And Weaknesses:**

Strengths
+ The authors propose a novel intrinsic sliced Wasserstein (ISW) for collections of probability measures by leveraging eigenfunctions for 1-dimensional projection and eigenvalues as weights.
+ The paper is easy to follow.
+ The authors derive hypothesis testing in the Hilbert for collections of probability measures problems under ISW.

Weaknesses
+ The ideas to use eigenfunctions and eigenvalues for sliced Wasserstein are interesting. However, the authors have not demonstrated the advantages of the proposed approach over other approaches (e.g., sliced-Wasserstein, general sliced-Wasserstein, tree-sliced-Wasserstein or the recent Sobolev transport – a variant of optimal transport on graphs. Note that all these approaches are Hilbert embeddable as the proposed method) in theory and/or experiments.
+ The authors should discuss the complexity of the proposed intrinsic sliced Wasserstein (and report time consumption in experiments)

---

> ### Author Response · Authors · 2022-08-02
> **Comparisons and details, limitations, and potential impacts**
>
> Thank you for your detailed comments. Please find our responses below.
>
> **Motivation of using eigenfunctions**
>
> The metric on the underlying space $\mathcal{X}$ has an important impact on the derived quantities: metrics that capture the geometry of the underlying space as relevant to the application would presumably lead to more successful derived quantities (see the 2nd point in response to Reviewer ib9Y). The use of eigenfunctions can be traced to such successful metrics on graphs and manifolds, such as Diffusion Distance useful in high-dimensional data settings and Biharmonic Distance useful for graphics applications (i.e. 2D manifolds).
>
> Additionally, eigenfunctions have a frequency structure like the usual sine/cosine basis in harmonic analysis on the periodic interval. Thus, e.g. in our testing framework the eigenfunctions can capture differences that are present at multiple resolution scales. There is quite a bit of literature on manifold learning methods that successfully employ Laplacian eigenfunctions, e.g. Laplacian Eigenmaps and follow-up works from Mikhail Belkin and others.
>
> On the theoretical side, we are able to use the classical results such as Weyl's eigenvalue laws and Hormander's bound on eigenfunction norms to prove the finiteness of the proposed distances. Eignefunctions/eigenvectors provide a common ground for building theory both on graphs and manifolds. None of the previous methods are able to capture both of these domains within a single elegant theory.
>
> **Comparison with SW and its variants**
>
> We first stress that on a theoretical level, none of the existing approaches provide a unified treatment for both graphs and manifolds. Figure 2b provides one such comparison for the finite interval case. Here, "unsliced" corresponds to the original 2-Wassestein distance (SW with a single possible projection in 1D case). As we can see ISW does better for $L=2,3$.
>
> We devised the following example to showcase the importance of the intrinsic definition of distance, and include experimental results in the comment below. Consider a finite 1D interval AB that is bent into a circle on the plane, where A and B are very close to each other but they do not touch. The intrinsic distance between A and B is large as it requires travel from one end of the interval to the other, in contrast the extrinsic Euclidean distance is small. No matter what Euclidean projection we use, the SW cost of moving probability mass from A to B would be small. This will lead to testing power loss when we compare two distribution sets that concentrate one around A and the other around B.
>
> **Relation between $T$ in Eq 2.1 and in Proposition 1**
>
> These quantities are the same. Eq. 2.1 gives the definition, and Proposition 1 establishes an equivalent formula. We will make this more explicit by using "equals by definition" notation in Eq 2.1.
>
> **Does Proposition 6 hold for any kernel used in MMD?**
>
> The kernel used for MMD should be the same as the one used for ISW; we shall state this more clearly.
>
> **In Proposition 8, why can one assume that C is finite?**
>
> On graphs, $C$ is trivially finite. On manifolds finiteness depends on the choice of the eigenfunction weighting function $\alpha(\cdot)$, and $C$ would be finite if $\alpha(\cdot)$ decays fast enough. For example this holds when $\alpha(\lambda)=e^{-t\lambda}$ for any $t>0$. Other choices can be derived by using the Weyl's law for eigenvalues (See Remark 1 in the Appendix for a relevant discussion).
>
> **For line 266-269, how to choose $L, D'$ to obtain infinitesimally small error estimation?**
>
> Please see Proposition 10 in the Appendix for the relevant statement.
>
> **For experiments choice of $L, D'$ vs. results in Section 3.2 and Section 4?**
>
> The choice of L and D' in the Proposition 10 relates to asymptotic results that highlight consistency and power characteristics of the
> proposed approach. However, in practice instead of asymptotics we use resampling null distribution for the test based on $\mathbb{T}(\cdot,\cdot)$.
>
> **Limitations**
>
> There is scope of exploration for choosing the parameters $L,D'$ in a principled manner. Empirical computation of the eigenfunctions for general manifolds will introduce approximation errors that need to be tackled by expanding the theoretical results. We plan to investigate these directions in future work.
>
> **Potential negative societal impacts**
>
> In privacy-sensitive situations, e.g. analyzing manifold-valued personal data, there could be privacy risks associated with releasing the $ISW_2$ embeddings vectors. Additional studies are warranted to quantify such risks and generate differentially private $ISW_2$ embeddings. Any difference between data from different demographic groups found using our procedure should be evaluated in light of potential biases stemming from the data collection procedure.

---

> > ### Author Response · Authors · 2022-08-06
> > **Experimental comparison with existing slicing techniques**
> >
> > We ran experiments using the setup in our earlier comment to compare $ISW_2$ with existing slicing-based techniques. We consider a planar regular 200-gon as a graph, except that one of the edges is removed. Denote the two degree-1 vertices as A and B. So basically, this is a path graph from A to B. Vertices A and B are located nearby each other on the plane but they do not have an edge connecting them. We generate distributions on this graph by putting Unif(0,1) weights at each of the vertices, and added an additional weight of 10 to the bin at point A to obtain the first set of distributions with a mode at A; For the second set we put the weight of 10 at B instead to obtain a set of distributions with mode at B. Following experimental settings in the main paper, we set $n_1=60,n_2=40$, and number of replications at 1000, to compute power and size of the hypothesis tests.
> >
> > The following table shows comparison of our method with SW in 2D, GSW with 3 choices of defining functions (circular, homogenous polynomial of degree 3, and degree 5), and Sobolev transport. ISW, SW, and GSW embeddings are done using 10 slices, and 8 inverse CDF mappings, i.e. $L=10, D'=8$ in the notation of the paper.
> >
> > | Method | Power | Size | Total time for 1000 replications(sec) |
> > | --- | --- | --- | --- |
> > | SW | 0.128 | 0.029 | 579.3 |
> > | GSW-circ |  0.128 | 0.029 | 564.82 |
> > | GSW-poly3 | 0.025 | 0.025 | 779.69 |
> > | GSW-poly5 | 0.044 | 0.035 | 1009.85 |
> > | ISW | 1.00 | 0.029 | 630.67 |
> > | Sobolev | 1.00 | 0.048 | 21474.88 |
> >
> > ISW achieves the best performance in terms of power, maintains nominal size (should be $\leq 0.05$), and has comparable computational time as SW/GSW that use the same number of slices. Sobolev transport achieves high power as well, but takes much longer than our method. This is because we use the $p$-value combination tests on the embeddings generated by other methods that is much faster, whereas for Sobolev transport we have to rely on permutation tests since it doesn't generate any embeddings.
> >
> > We were not able to adopt/implement Tree-sliced Wasserstein due to the relative complexity of the [MATLAB code](https://github.com/lttam/TreeWasserstein) and not having a MATLAB license ourselves. However, given the extensive array of experimental settings and diversity of competing methods we have considered in the experiments in the main paper and this rebuttal, we believe the effectiveness of our proposed intrinsic slicing technique is evident.

---

### Meta-Review · Area_Chair_Wq43 · 2022-08-24

**Recommendation:** Reject
**Confidence:** Certain

**Metareview:**

In this paper, the authors propose the intrinsic sliced Wasserstein distances and a Hypothesis testing framework for the proposed measure. The idea of using eigenfunctions and eigenvalues for sliced Wasserstein is interesting. The authors addressed a part of the concerns raised by the reviewers. However, the advantage over the existing methods (sliced Wasserstein distances and MMDs) is unclear. Thus, it needs a major revision and cannot be accepted with the current version. I encourage the authors to revise the paper based on the reviewer's comments and resubmit it to a future venue.

**Award:**

No

---

### Decision · Program_Chairs · 2022-09-14

Reject